# Empowering Graph Representation Learning with Test-Time Graph Transformation

**Wei Jin**[1][*] **Tong Zhao**[2] **, Jiayuan Ding**[1] **, Yozen Liu**[2] **, Jiliang Tang**[1] **and Neil Shah**[2]
[1]Michigan State University   [2]Snap Inc.
{jinwei2,dingjia5,tangjili}@msu.edu, {tzhao,yliu2,nshah}@snap.com

## ABSTRACT

As powerful tools for representation learning on graphs, graph neural networks (GNNs) have facilitated various applications from drug discovery to recommender systems. Nevertheless, the effectiveness of GNNs is immensely challenged by issues related to data quality, such as distribution shift, abnormal features and adversarial attacks. Recent efforts have been made on tackling these issues from a modeling perspective which requires additional cost of changing model architectures or re-training model parameters. In this work, we provide a data-centric view to tackle these issues and propose a graph transformation framework named GTRANS which adapts and refines graph data at test time to achieve better performance. We provide theoretical analysis on the design of the framework and discuss why adapting graph data works better than adapting the model. Extensive experiments have demonstrated the effectiveness of GTRANS on three distinct scenarios for eight benchmark datasets where suboptimal data is presented. Remarkably, GTRANS performs the best in most cases with improvements up to 2.8%, 8.2% and 3.8% over the best baselines on three experimental settings. Code is released at https://github.com/ChandlerBang/GTrans.

## 1 INTRODUCTION

Graph representation learning has been at the center of various real-world applications, such as drug discovery (Duvenaud et al., 2015; Guo et al., 2022), recommender systems (Ying et al., 2018; Fan et al., 2019; Sankar et al., 2021), forecasting (Tang et al., 2020; Derrow-Pinion et al., 2021) and outlier detection (Zhao et al., 2021a; Deng & Hooi, 2021). In recent years, there has been a surge of interest in developing graph neural networks (GNNs) as powerful tools for graph representation learning (Kipf & Welling, 2016a; Veličković et al., 2018; Hamilton et al., 2017; Wu et al., 2019). Remarkably, GNNs have achieved state-of-the-art performance on numerous graph-related tasks including node classification, graph classification and link prediction (Chien et al., 2021; You et al., 2021; Zhao et al., 2022b).

Despite the enormous success of GNNs, recent studies have revealed that their generalization and robustness are immensely challenged by the data quality (Jin et al., 2021b; Li et al., 2022). In particular, GNNs can behave unreliably in scenarios where sub-optimal data is presented:

1. *Distribution shift* (Wu et al., 2022a; Zhu et al., 2021a). GNNs tend to yield inferior performance when the distributions of training and test data are not aligned (due to corruption or inconsistent collection procedure of test data).
2. *Abnormal features* (Liu et al., 2021a). GNNs suffer from high classification errors when data contains abnormal features, e.g., incorrect user profile information in social networks.
3. *Adversarial structure attack* (Zügner et al., 2018; Li et al., 2021). GNNs are vulnerable to imperceptible perturbations on the graph structure which can lead to severe performance degradation.

To tackle these problems, significant efforts have been made on developing new techniques from the *modeling perspective*, e.g., designing new architectures and employing adversarial training strategies (Xu et al., 2019; Wu et al., 2022a). However, employing these methods in practice may be

---

[*]Work done while author was on internship at Snap Inc.

Figure 1: We study the test-time graph transformation problem, which seeks to learn a refined graph such that pre-trained GNNs can perform better on the new graph compared to the original. **Shown:** An illustration of our proposed approach's empirical performance on transforming a noisy graph.

infeasible, as they require additional cost of changing model architectures or re-training model parameters, especially for well-trained large-scale models. The problem is further exacerbated when adopting these techniques for multiple architectures. By contrast, this paper seeks to investigate approaches that can be readily used with a wide variety of pre-trained models and test settings for improving model generalization and robustness. Essentially, we provide a *data-centric perspective* to address the aforementioned issues by modifying the graph data presented at test-time. Such modification aims to bridge the gap between training data and test data, and thus enable GNNs to achieve better generalization and robust performance on the new graph. Figure 1 visually describes this idea: we are originally given with a test graph with abnormal features where multiple GNN architectures yield poor performance; however, by transforming the graph prior to inference (at test-time), we enable these GNNs to achieve much higher accuracy.

In this work, we aim to develop a data-centric framework that transforms the test graph to enhance model generalization and robustness, without altering the pre-trained model. In essence, we are faced with two challenges: (1) how to model and optimize the transformed graph data, and (2) how to formulate an objective that can guide the transformation process. First, we model the graph transformation as injecting perturbation on the node features and graph structure, and optimize them alternatively via gradient descent. Second, inspired by the recent progress of contrastive learning, we propose a parameter-free surrogate loss which does not affect the pre-training process while effectively guiding the graph adaptation. Our contributions can be summarized as follows:

1. For the first time, we provide a data-centric perspective to improve the generalization and robustness of GNNs with test-time graph transformation.
2. We establish a novel framework GTRANS for test-time graph transformation by jointly learning the features and adjacency structure to minimize a proposed surrogate loss.
3. Our theoretical analysis provides insights on what surrogate losses we should use during test-time graph transformation and sheds light on the power of data-adaptation over model-adaptation.
4. Extensive experimental results on three settings (distribution shift, abnormal features and adversarial structure attacks) have demonstrated the superiority of test-time graph transformation. Particularly, GTRANS performs the best in most cases with improvements up to 2.8%, 8.2% and 3.8% over the best baselines on three experimental settings.

Moreover, we note: (1) GTRANS is flexible and versatile. It can be equipped with any pre-trained GNNs and the outcome (the refined graph data) can be deployed with any model given its favorable transferability. (2) GTRANS provides a degree of interpretability, as it can show which kinds of graph modifications can help improve performance by visualizing the data.

## 2    RELATED WORK

**Distribution shift in GNNs.** GNNs have revolutionized graph representation learning and achieved state-of-the-art results on diverse graph-related tasks (Kipf & Welling, 2016a; Veličković et al., 2018; Hamilton et al., 2017; Chien et al., 2021; Klicpera et al., 2018; Wu et al., 2019). However, recent studies have demonstrated that GNNs yield sub-optimal performance on out-of-distribution data for node classification (Zhu et al., 2021a; Wu et al., 2022a; Liu et al., 2022a) and graph classification (Chen et al., 2022; Buffelli et al., 2022; Gui et al., 2022; Wu et al., 2022b; You et al., 2023). These studies have introduced solutions to tackle distribution shifts by altering model training behavior or model architectures. For a thorough review, we refer the readers to a recent survey (Li et al., 2022). Unlike existing works, we target modifying the inputs via test-time adaption.

**Robustness of GNNs.** Recent studies have demonstrated the vulnerability of GNNs to graph adversarial attacks (Zügner et al., 2018; Zügner & Günnemann, 2019; Xu et al., 2019; Geisler et al.,

2021), i.e., small perturbations on the input graph can mislead GNNs into making wrong predictions. Several works make efforts towards developing new GNNs or adversarial training strategies to defend against attacks (Xu et al., 2019; Zhu et al., 2019; Jin et al., 2021a;b). Instead of altering model training behavior, our work aims to modify the test graph to correct adversarial patterns.

**Graph Structure Learning & Graph Data Augmentation.** Graph structure learning and graph data augmentation both aim to improve GNNs' generalization performance by augmenting the (training) graph data (Zhao et al., 2022a), either learning the graph from scratch (Franceschi et al., 2019; Jin et al., 2020; Chen et al., 2020; Zhao et al., 2021b) or perturbing the graph in a rule-based way (Rong et al., 2020; Feng et al., 2020; Ding et al., 2022). While our work also modifies the graph data, we focus on modifying the test data and not impacting the model training process.

**Test-time Training.** Our work is also related to test-time training (Sun et al., 2020; Wang et al., 2021; Liu et al., 2021b; Zhang et al., 2021; 2022), which has raised a surge of interest in computer vision recently. To handle out-of-distribution data, Sun et al. (2020) propose the pioneer work of test-time training (TTT) by optimizing feature extractor via an auxiliary task loss. However, TTT alters training to jointly optimize the supervised loss and auxiliary task loss. To remove the need for training an auxiliary task, Tent (Wang et al., 2021) proposes to minimize the prediction entropy at test-time. Tent works by adapting the parameters in batch normalization layers, which may not always be employed by modern GNNs. In this work, we focus on a novel perspective of adapting the test graph data, which makes no assumptions about the particular training procedure or architecture.

## 3 METHODOLOGY

We start by introducing the general problem of test-time graph transformation (TTGT). While our discussion mainly focuses on the node classification task where the goal is to predict the labels of nodes in the graph, it can be easily extended to other tasks. Consider that we have a training graph $G_{Tr}$ and a test graph $G_{Te}$, and the corresponding set of node labels are denoted as $\mathcal{Y}_{Tr}$ and $\mathcal{Y}_{Te}$, respectively. Note that the node sets in $G_{Tr}$ and $G_{Te}$ can either be disjoint or overlapping, and they are not necessarily drawn from the same distribution. Further, let $f_\theta(\cdot)$ denote the mapping function of a GNN model parameterized by $\theta$, which maps a graph into the space of node labels.

**Definition 1** (Test-Time Graph Transformation (TTGT)). *TTGT requires to learn a graph transformation function $g(\cdot)$ which refines the test graph $G_{Te}$ such that the pre-trained $f_\theta$ can yield better test performance on $g(G_{Te})$ than that on $G_{Te}$:*

$$\arg \min_g \mathcal{L}\left(f_{\theta^*}(g(G_{Te})), \mathcal{Y}_{Te}\right) \quad s.t. \quad g(G_{Te}) \in \mathcal{P}(G_{Te}),$$
$$\text{with } \theta^* = \arg \min_\theta \mathcal{L}\left(f_\theta(G_{Tr}), \mathcal{Y}_{Tr}\right),$$

(1)

*where $\mathcal{L}$ denotes the loss function measuring downstream performance; $\mathcal{P}(G_{Te})$ is the space of the modified graph, e.g., we may constrain the change on the graph to be small.*

To optimize the TTGT problem, we are faced with two critical challenges: (1) how to parameterize and optimize the graph transformation function $g(\cdot)$; and (2) how to formulate a surrogate loss to guide the learning process, since we do not have access to the ground-truth labels of test nodes. Therefore, we propose GTRANS and elaborate on how it addresses these challenges as follows.

### 3.1 CONSTRUCTING GRAPH TRANSFORMATION

Let $G_{Te} = \{\mathbf{A}, \mathbf{X}\}$ denote the test graph, where $\mathbf{A} \in \{0, 1\}^{N \times N}$ is the adjacency matrix, $N$ is the number of nodes, and $\mathbf{X} \in \mathbb{R}^{N \times d}$ is the $d$-dimensional node feature matrix. Since the pre-trained GNN parameters are fixed at test time and we only care about the test graph, we drop the subscript in $G_{Te}$ and $\mathcal{Y}_{Te}$ to simplify notations in the rest of the paper.

**Construction.** We are interested in obtaining the transformed test graph $G'_{Te} = g(\mathbf{A}, \mathbf{X}) = (\mathbf{A}', \mathbf{X}')$. Specifically, we model feature modification as an additive function which injects perturbation to node features, i.e., $\mathbf{X}' = \mathbf{X} + \Delta_{\mathbf{X}}$; we model the structure modification as $\mathbf{A}' = \mathbf{A} \oplus \Delta_{\mathbf{A}}$[1], where $\oplus$ stands for an element-wise exclusive OR operation and $\Delta_{\mathbf{A}} \in \{0, 1\}^{N \times N}$ is a binary matrix. In other words, $(\Delta_{\mathbf{A}})_{ij} = 1$ indicates an edge flip. Formally, we seek to find $\Delta_{\mathbf{A}}$ and $\Delta_{\mathbf{X}}$ that

---

[1] $(\mathbf{A} \oplus \Delta_{\mathbf{A}})_{ij}$ can be implemented as $2 - (\mathbf{A} + \Delta_{\mathbf{A}})_{ij}$ if $(\mathbf{A} + \Delta_{\mathbf{A}})_{ij} \geq 1$, otherwise $(\mathbf{A} + \Delta_{\mathbf{A}})_{ij}$.

can minimize the objective function:

$$\underset{\Delta_\mathbf{A}, \Delta_\mathbf{X}}{\arg\min} \mathcal{L}\left(f_\theta(\mathbf{A} \oplus \Delta_\mathbf{A}, \mathbf{X} + \Delta_\mathbf{X}), \mathcal{Y}\right) \quad \text{s.t.} \quad (\mathbf{A} \oplus \Delta_\mathbf{A}, \mathbf{X} + \Delta_\mathbf{X}) \in \mathcal{P}(\mathbf{A}, \mathbf{X}), \qquad (2)$$

where $\Delta_\mathbf{A} \in \{0, 1\}^{N \times N}$ and $\Delta_\mathbf{X} \in \mathbb{R}^{N \times d}$ are treated as free parameters. Further, to ensure we do not heavily violate the original graph structure, we constrain the number of changed entries in the adjacency matrix to be smaller than a budget $B$ on the graph structure, i.e., $\sum \Delta_\mathbf{A} \leq B$. We do not impose constraints on the node features to ease optimization. In this context, $\mathcal{P}$ can be viewed as constraining $\Delta_\mathbf{A}$ to a binary space as well as restricting the number of changes.

**Optimization.** The optimization for $\Delta_\mathbf{X}$ is easy since the node features are continuous. The optimization for $\Delta_\mathbf{A}$ is particularly difficult in that (1) $\Delta_\mathbf{A}$ is binary and constrained; and (2) the search space of $N^2$ entries is too large especially when we are learning on large-scale graphs.

To cope with the first challenge, we relax the binary space to $[0, 1]^{N \times N}$ and then we can employ projected gradient descent (PGD) (Xu et al., 2019; Geisler et al., 2021) to update $\Delta_\mathbf{A}$:

$$\Delta_\mathbf{A} \leftarrow \Pi_\mathcal{P}\left(\Delta_\mathbf{A} - \eta \nabla_{\Delta_\mathbf{A}} \mathcal{L}(\Delta_\mathbf{A})\right) \qquad (3)$$

where we first perform gradient descent with step size $\eta$ and call a projection $\Pi_\mathcal{P}$ to project the variable to the space $\mathcal{P}$. Specifically, given an input vector $\mathbf{p}$, $\Pi_\mathcal{P}(\cdot)$ is expressed as:

$$\Pi_\mathcal{P}(\mathbf{p}) \leftarrow \begin{cases} \Pi_{[0,1]}(\mathbf{p}), & \text{If } \mathbf{1}^\top \Pi_{[0,1]}(\mathbf{p}) \leq B; \\ \Pi_{[0,1]}(\mathbf{p} - \gamma \mathbf{1}) \text{ with } \mathbf{1}^\top \Pi_{[0,1]}(\mathbf{p} - \gamma \mathbf{1}) = B, & \text{otherwise,} \end{cases} \qquad (4)$$

where $\Pi_{[0,1]}(\cdot)$ clamps the input values to $[0, 1]$, $\mathbf{1}$ stands for a vector of all ones, and $\gamma$ is obtained by solving the equation $\mathbf{1}^\top \Pi_{[0,1]}(\mathbf{p} - \gamma \mathbf{1}) = B$ with the bisection method (Liu et al., 2015). To keep the adjacency structure discrete and sparse, we view each entry in $\mathbf{A} \oplus \Delta_\mathbf{A}$ as a Bernoulli distribution and sample the learned graph as $\mathbf{A}' \sim \text{Bernoulli}(\mathbf{A} \oplus \Delta_\mathbf{A})$.

Furthermore, to enable efficient graph structure learning, it is desired to reduce the search space of updated adjacency matrix. One recent approach of graph adversarial attack (Geisler et al., 2021) proposes to sample a small block of possible entries from the adjacency matrix and update them at each iteration. This solution is still computationally intensive as it requires hundreds of steps to achieve a good performance. Instead, we constrain the search space to only the existing edges of the graph, which is typically sparse. Empirically, we observe that this simpler strategy still learns useful structure information when combined with feature modification.

## 3.2 Parameter-Free Surrogate Loss

As discussed earlier, the proposed framework GTRANS aims to improve the model generalization and robustness by learning to transform the test graph. Ideally, when we have test ground-truth labels, the problem can be readily solved by adapting the graph to result in the minimum cross entropy loss on test samples. However, as we do not have such information at test-time, it motivates us to investigate feasible surrogate losses to guide the graph transformation process. In the absence of labeled data, recently emerging self-supervised learning techniques (Xie et al., 2021; Liu et al., 2022b) have paved the way for providing self-supervision for TTGT. However, not every surrogate self-supervised task and loss is suitable for our transformation process, as some tasks are more powerful and some are weaker. To choose a suitable surrogate loss, we provide the following theorem.

**Theorem 1.** *Let $\mathcal{L}_c$ denote the classification loss and $\mathcal{L}_s$ denote the surrogate loss, respectively. Let $\rho(G)$ denote the correlation between $\nabla_G \mathcal{L}_c(G, \mathcal{Y})$ and $\nabla_G \mathcal{L}_s(G)$, and let $\epsilon$ denote the learning rate for gradient descent. Assume that $\mathcal{L}_c$ is twice-differentiable and its Hessian matrix satisfies $\|\mathbf{H}(G, \mathcal{Y})\|_2 \leq M$ for all $G$. When $\rho(G) > 0$ and $\epsilon < \frac{2\rho(G)\|\nabla_G \mathcal{L}_c(G, \mathcal{Y})\|_2}{M \|\nabla_G \mathcal{L}_s(G)\|_2}$, we have*

$$\mathcal{L}_c\left(G - \epsilon \nabla_G \mathcal{L}_s(G), \mathcal{Y}\right) < \mathcal{L}_c(G, \mathcal{Y}). \qquad (5)$$

The proof can be found in Appendix A.1. Theorem 1 suggests that when the gradients from classification loss and surrogate loss have a positive correlation, i.e., $\rho(G) > 0$, we can update the test graph by performing gradient descent with a sufficiently small learning rate such that the classification loss on the test samples is reduced. Hence, it is imperative to find a surrogate task that shares relevant information with the classification task. To empirically verify the effectiveness of Theorem 1, we adopt the surrogate loss in Equation (6) as $\mathcal{L}_s$ and plot the values of $\rho(G)$ and $\mathcal{L}_c$ on one test graph in Cora in Figure 2. We can observe that a positive $\rho(G)$ generally reduces the test loss. Results on different losses can be found in Appendix D.9 and similar patterns are exhibited.

**Parameter-Free Surrogate Loss.** As one popular self-supervised paradigm, graph contrastive learning has achieved promising performance in various tasks (Hassani & Khasahmadi, 2020; You et al., 2021; Zhu et al., 2021b), which indicates that graph contrastive learning tasks are often highly correlated with downstream tasks. This property is desirable for guiding TTGT as suggested by Theorem 1. At its core lies the contrasting scheme, where the similarity between two augmented views from the same sample is maximized, while the similarity between

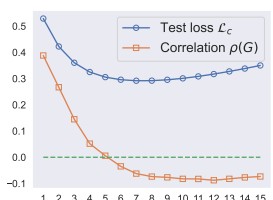

Figure 2: Positive $\rho(G)$ can help reduce test loss.

views from two different samples is minimized. However, the majority of existing graph contrastive learning methods cannot be directly applied to our scenario, as they often require a parameterized projection head to map augmented representations to another latent space, which inevitably alters the model architecture. Thus, we design a parameter-free surrogate loss which removes the projection head. Specifically, we apply an augmentation function $\mathcal{A}(\cdot)$ on input graph $G$ and obtain an augmented graph $\mathcal{A}(G)$. The node representations obtained from the two graphs are denoted as $Z$ and $\hat{Z}$, respectively; $\mathbf{z}_i$ and $\hat{\mathbf{z}}_i$ stand for the $i$-th node representation taken from them, respectively. We adopt DropEdge (Rong et al., 2020) as the augmentation function $\mathcal{A}(\cdot)$, and the node representations are taken from the second last layer of the trained model. Essentially, we maximize the similarity between original nodes and their augmented view while penalizing the similarity between the nodes and their negative samples:

$$\min \mathcal{L}_s = \sum_{i=1}^{N}(1 - \frac{\hat{\mathbf{z}}_i^\top \mathbf{z}_i}{\|\hat{\mathbf{z}}_i\|\|\mathbf{z}_i\|}) - \sum_{i=1}^{N}(1 - \frac{\tilde{\mathbf{z}}_i^\top \mathbf{z}_i}{\|\tilde{\mathbf{z}}_i\|\|\mathbf{z}_i\|}), \qquad (6)$$

where $\{\tilde{\mathbf{z}}_i | i = 1, \dots, N\}$ are the negative samples for corresponding nodes, which are generated by shuffling node features (Velickovic et al., 2019). In Eq. (6), the first term encourages each node to be close while the second term pushes each node away from the corresponding negative sample. Note that (1) $\mathcal{L}_s$ is parameter-free and does not require modification of the model architecture, or affect the pre-training process; (2) there could be other self-supervised signals for guiding the graph transformation, and we empirically compare them with the contrastive loss in Appendix D.9. We also highlight that our unique contribution is not the loss in Eq. (6) but the proposed TTGT framework as well as the theoretical and empirical insights on how to choose a suitable surrogate loss. Furthermore, the algorithm of GTRANS is provided in Appendix B.

### 3.3 FURTHER ANALYSIS

In this subsection, we study the theoretical property of Eq. (6) and compare the strategy of adapting data versus that of adapting model. We first demonstrate the rationality of the proposed surrogate loss through the following theorem.

**Theorem 2.** *Assume that the augmentation function $\mathcal{A}(\cdot)$ generates a data view of the same class for the test nodes and the node classes are balanced. Assume for each class, the mean of the representations obtained from $Z$ and $\hat{Z}$ are the same. Minimizing the first term in Eq. (6) is approximately minimizing the class-conditional entropy $H(Z|Y)$ between features $Z$ and labels $Y$.*

The proof can be found in Appendix A.2. Theorem 2 indicates that minimizing the first term in Eq. (6) will approximately minimize $H(Z|Y)$, which encourages high intra-class compactness, i.e., learning a low-entropy cluster in the embedded space for each class. Notably, $H(Z|Y)$ can be rewritten as $H(Z|Y) = H(Z) - I(Z, Y)$. It indicates that minimizing Eq. (6) can also help promote $I(Z, Y)$, the mutual information between the hidden representation and downstream class. However, we note that only optimizing this term can result in collapse (mapping all data points to a single point in the embedded space), which stresses the necessity of the second term in Eq. (6).

Next, we use an illustrative example to show that adapting data at test-time can be more useful than adapting model in some cases. Given test samples $\{\mathbf{x}_i | i = 1, \dots, K\}$, we consider a linearized GNN $f_\theta$ which first performs aggregation through a function $\text{Agg}(\cdot, \cdot)$ and then transforms the aggregated features via a function $\text{Trans}(\cdot)$. Hence, only the function $\text{Trans}(\cdot)$ is parameterized by $\theta$.

**Example.** Let $\mathcal{N}_i$ denote the neighbors for node $\mathbf{x}_i$. If there exist two nodes with the same aggregated features but different labels, i.e., $\text{Agg}(\mathbf{x}_1, \{\mathbf{x}_i | i \in \mathcal{N}_1\}) = \text{Agg}(\mathbf{x}_2, \{\mathbf{x}_j | j \in \mathcal{N}_2\})$, $y_1 \neq y_2$, adapting the data $\{\mathbf{x}_i | i = 1, \dots, K\}$ can achieve lower classification error than adapting the model $f_\theta$ at test stage.

*Illustration.* Let $\bar{\mathbf{x}}_1 = \text{Agg}(\mathbf{x}_1, \{\mathbf{x}_i | i \in \mathcal{N}_1\})$ and $\bar{\mathbf{x}}_2 = \text{Agg}(\mathbf{x}_2, \{\mathbf{x}_j | j \in \mathcal{N}_2\})$. For simplicity, we

consider the following mean square loss as the classification error:

$$\ell = \frac{1}{2}\left(\text{Trans}(\bar{\mathbf{x}}_1)) - y_1)^2 + (\text{Trans}(\bar{\mathbf{x}}_2) - y_2)^2\right). \tag{7}$$

It is easy to see that $\ell$ reaches its minimum when $\text{Trans}(\bar{\mathbf{x}}_1) = y_1$ and $\text{Trans}(\bar{\mathbf{x}}_2) = y_2$. In this context, it is impossible to find $\theta$ such that $\text{Trans}(\cdot)$ can map $\mathbf{x}_1, \mathbf{x}_2$ to different labels since it is not a one-to-many function. However, since $y_1$ and $y_2$ are in the label space of training data, we can always modify the test graph to obtain newly aggregated features $\bar{\mathbf{x}}'_1, \bar{\mathbf{x}}'_2$ such that $\text{Trans}(\bar{\mathbf{x}}'_1) = y_1$ and $\text{Trans}(\bar{\mathbf{x}}'_2) = y_2$, which minimizes $\ell$. In the extreme case, we may drop all node connections for the two nodes, and let $\mathbf{x}_1 \leftarrow \bar{\mathbf{x}}'_1$ and $\mathbf{x}_2 \leftarrow \bar{\mathbf{x}}'_2$ where $\bar{\mathbf{x}}'_1$ and $\bar{\mathbf{x}}'_2$ are the aggregated features taken from the training set. Hence, adapting data can achieve lower classification loss.

**Remark 1.** Note that the existence of two nodes with the same aggregated features but different labels is not rare when considering adversarial attack or abnormal features. We provide a figurative example in Figure 6 in Appendix A.3: the attacker injects one adversarial edge into the graph and changes the aggregated features $\bar{\mathbf{x}}_1$ and $\bar{\mathbf{x}}_2$ to be the same.

**Remark 2.** When we consider $\bar{\mathbf{x}}_1 \neq \bar{\mathbf{x}}_2, y_1 \neq y_2$, whether we can find $\theta$ satisfying $\text{Trans}(\bar{\mathbf{x}}_1) = y_1$ and $\text{Trans}(\bar{\mathbf{x}}_2) = y_2$ depends on the expressiveness of the the transformation function. If it is not powerful enough (e.g., an under-parameterized neural network), it could fail to map different data points to different labels. On the contrary, adapting the data does not suffer this problem as we can always modify the test graph to satisfy $\text{Trans}(\bar{\mathbf{x}}'_1) = y_1$ and $\text{Trans}(\bar{\mathbf{x}}'_2) = y_2$.

**Remark 3.** The above discussion can be easily extended to nonlinear GNN by considering $\bar{\mathbf{x}}_1, \bar{\mathbf{x}}_2$ as the output before the last linear layer of GNN.

## 4 EXPERIMENT

### 4.1 GENERALIZATION ON OUT-OF-DISTRIBUTION DATA

**Setup.** Following the settings in EERM (Wu et al., 2022a), which is designed for node-level tasks on OOD data, we validate GTRANS on three types of distribution shifts with six benchmark datasets: (1) artificial transformation for Cora (Yang et al., 2016) and Amazon-Photo (Shchur et al., 2018), (2) cross-domain transfers for Twitch-E and FB-100 (Rozemberczki et al., 2021a) (Lim et al., 2021), and (3) temporal evolution for Elliptic (Pareja et al., 2020) and OGB-Arxiv (Hu et al., 2020). Moreover, Cora and Amazon-Photo have 1/1/8 graphs for training/validation/test sets. The splits are 1/1/5 on Twitch-E, 3/2/3 on FB-100, 5/5/33 on Elliptic, and 1/1/3 on OGB-Arxiv. More details on the datasets are provided in Appendix C. We compare GTRANS with four baselines: empirical risk minimization (ERM, i.e., standard training), data augmentation technique DropEdge (Rong et al., 2020), test-time-training method Tent (Wang et al., 2021), and the recent SOTA method EERM (Wu et al., 2022a) which is exclusively developed for graph OOD issue. Further, we evaluate all the methods with four popular GNN backbones including GCN (Kipf & Welling, 2016a), GraphSAGE (Hamilton et al., 2017), GAT (Veličković et al., 2018), and GPR (Chien et al., 2021). Their default setup follows that in EERM[2]. We refer the readers to Appendix C.1 for more implementation details of baselines and GTRANS. Notably, all experiments in this paper are repeated 10 times with different random seeds. Due to page limit, we include more baselines such as SR-GNN (Zhu et al., 2021a) and UDA-GCN (Wu et al., 2020) in Appendix D.1.

**Results.** Table 1 reports the averaged performance over the test graphs for each dataset as well as the averaged rank of each algorithm. From the table, we make the following observations:

(a) Overall Performance. The proposed framework consistently achieves strong performance across the datasets: GTRANS achieves average ranks of 1.0, 1.7, 2.0 and 1.7 with GCN, SAGE, GAT and GPR, respectively, while the corresponding ranks for the best baseline EERM are 2.9, 3.4, 3.0 and 2.0. Furthermore, in most of the cases, GTRANS significantly improves the vanilla baseline (ERM) by a large margin. Particularly, when using GCN as backbone, GTRANS outperforms ERM by 3.1%, 5.0% and 2.0% on Cora, Elliptic and OGB-Arxiv, respectively. These results demonstrate the effectiveness of GTRANS in tackling diverse types of distribution shifts.

(b) Comparison to other baselines. Both DropEdge and EERM modify the training process to improve model generalization. Nonetheless, they are less effective than GTRANS, as GTRANS takes

---

[2]We note that the GCN used in the experiments of EERM does not normalize the adjacency matrix according to its open-source code. Here we normalize the adjacency matrix to make it consistent with the original GCN.

Table 1: Average classification performance (%) on the test graphs. Rank indicates the average rank of each algorithm for each backbone. OOM indicates out-of-memory error on 32 GB GPU memory. The proposed GTRANS consistently ranks the best compared with the baselines. */** indicates that GTrans outperforms ERM at the confidence level 0.1/0.05 from paired t-test.

| Backbone | Method | Amz-Photo | Cora | Elliptic | FB-100 | OGB-Arxiv | Twitch-E | Rank |
|---|---|---|---|---|---|---|---|---|
| GCN | ERM | 93.79±0.97 | 91.59±1.44 | 50.90±1.51 | 54.04±0.94 | 38.59±1.35 | 59.89±0.50 | 3.8 |
| | DropEdge | 92.11±0.31 | 81.01±1.33 | 53.96±4.91 | 53.00±0.50 | 41.26±0.92 | 59.95±0.39 | 3.6 |
| | Tent | 94.03±1.07 | 91.87±1.36 | 51.71±2.00 | 54.16±1.00 | 39.33±1.40 | 59.46±0.55 | 3.3 |
| | EERM | 94.05±0.40 | 87.21±0.53 | 53.96±0.65 | 54.24±0.55 | OOM | 59.85±0.85 | 2.9 |
| | GTRANS | **94.13±0.77*** | **94.66±0.63**** | **55.88±3.10**** | 54.32±0.60 | **41.59±1.20**** | **60.42±0.86*** | **1.0** |
| SAGE | ERM | 95.09±0.60 | 99.67±0.14 | 56.12±4.47 | 54.70±0.47 | 39.56±1.66 | 62.06±0.09 | 3.2 |
| | DropEdge | 92.61±0.56 | 95.85±0.30 | 52.38±3.11 | 54.51±0.69 | 38.89±1.74 | 62.14±0.12 | 4.2 |
| | Tent | 95.72±0.43 | **99.80±0.10** | 55.89±4.87 | **54.86±0.34** | 39.58±1.26 | 62.09±0.09 | 2.3 |
| | EERM | 95.57±0.13 | 98.77±0.14 | 58.20±3.55 | 54.28±0.97 | OOM | 62.11±0.12 | 3.4 |
| | GTRANS | **96.91±0.68**** | 99.45±0.13 | **60.81±5.19**** | 54.64±0.62 | **40.39±1.45**** | **62.15±0.13*** | **1.7** |
| GAT | ERM | 96.30±0.79 | 94.81±1.28 | 65.36±2.70 | 51.77±1.41 | 40.63±1.57 | 58.53±1.00 | 3.0 |
| | DropEdge | 90.70±0.29 | 76.91±1.55 | 63.78±2.39 | 52.65±0.88 | 42.48±0.93 | 58.89±1.01 | 3.3 |
| | Tent | 95.99±0.46 | 95.91±1.14 | 66.07±1.66 | 51.47±1.70 | 40.06±1.19 | 58.33±1.18 | 3.3 |
| | EERM | 95.57±1.32 | 85.00±0.96 | 58.14±4.71 | **53.30±0.77** | OOM | **59.84±0.71** | 3.0 |
| | GTRANS | **96.67±0.74**** | **96.37±1.00**** | **66.43±2.57**** | 51.16±1.72 | **43.76±1.25**** | 58.59±1.07 | **2.0** |
| GPR | ERM | 91.87±0.65 | 93.00±2.17 | 64.59±3.52 | 54.51±0.33 | 44.38±0.59 | 59.72±0.40 | 2.7 |
| | DropEdge | 88.81±1.48 | 79.27±1.39 | 61.02±1.78 | 55.04±0.33 | 43.65±0.77 | 59.89±0.05 | 3.3 |
| | Tent[3] | - | - | - | - | - | - | - |
| | EERM | 90.78±0.52 | 88.82±3.10 | 67.27±0.98 | **55.95±0.03** | OOM | **61.57±0.12** | 2.0 |
| | GTRANS | **91.93±0.73** | 93.05±2.02 | **69.03±2.33**** | 54.38±0.31 | **46.00±0.46**** | 60.11±0.53** | **1.7** |

[3] Tent cannot be applied to models which do not contain batch normalization layers.

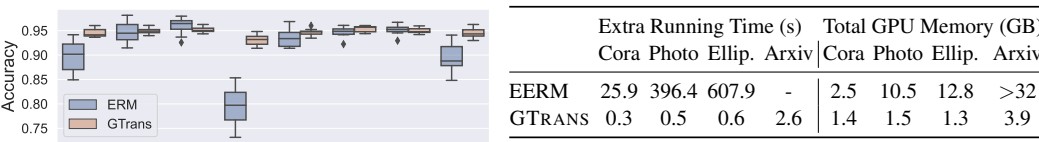

| | Extra Running Time (s) | | | | Total GPU Memory (GB) | | | |
|---|---|---|---|---|---|---|---|---|
| | Cora | Photo | Ellip. | Arxiv | Cora | Photo | Ellip. | Arxiv |
| EERM | 25.9 | 396.4 | 607.9 | - | 2.5 | 10.5 | 12.8 | >32 |
| GTRANS | 0.3 | 0.5 | 0.6 | 2.6 | 1.4 | 1.5 | 1.3 | 3.9 |

Figure 3: Results on Cora under OOD. GTRANS improves GCN on most test graphs.

Table 2: Efficiency comparison. GTRANS is more time- and memory-efficient than EERM.

advantage of the information from test graphs. As a test-time training method, Tent also performs well in some cases, but Tent only adapts the parameters in batch normalization layers and cannot be applied to models without batch normalization.

We further show the performance on each test graph on Cora with GCN in Figure 3 and the results for other datasets are provided in Appendix D.4. We observe that GTRANS generally improves over individual test graphs within each dataset, which validates the effectiveness of GTRANS.

**Efficiency Comparison.** Since EERM performs the best among baselines, Table 2 showcases the efficiency comparison between our proposed GTRANS and EERM on the largest test graph in each dataset. The additional running time of GTRANS majorly depends on the number of gradient descent steps. As we only use a small number (5 or 10) throughout all the experiments, the time overhead brought by GTRANS is negligible. Compared with the re-training method EERM, GTRANS avoids the complex bilevel optimization and thus is significantly more efficient. Furthermore, EERM imposes a considerably heavier memory burden.

## 4.2 ROBUSTNESS TO ABNORMAL FEATURES

**Setup.** Following the setup in AirGNN (Liu et al., 2021a), we evaluate the robustness in the case of abnormal features. Specifically, we simulate abnormal features by assigning random features taken from a multivariate standard Gaussian distribution to a portion of randomly selected test nodes. Note that the abnormal features are injected after model training (at test time) and we vary the ratio of noisy nodes from 0.1 to 0.4 with a step size of 0.05. This process is performed for four datasets: the original version of Cora, Citeseer, Pubmed, and OGB-Arxiv. In these four datasets, the training graph and the test graph have the same graph structure but the node features are different. Hence, we use the training classification loss combined with the proposed contrastive loss to optimize GTRANS. We use GCN as the backbone model and adopt four GNNs as the baselines including GAT (Veličković et al., 2018), APPNP (Klicpera et al., 2018), AirGNN and AirGNN-t.

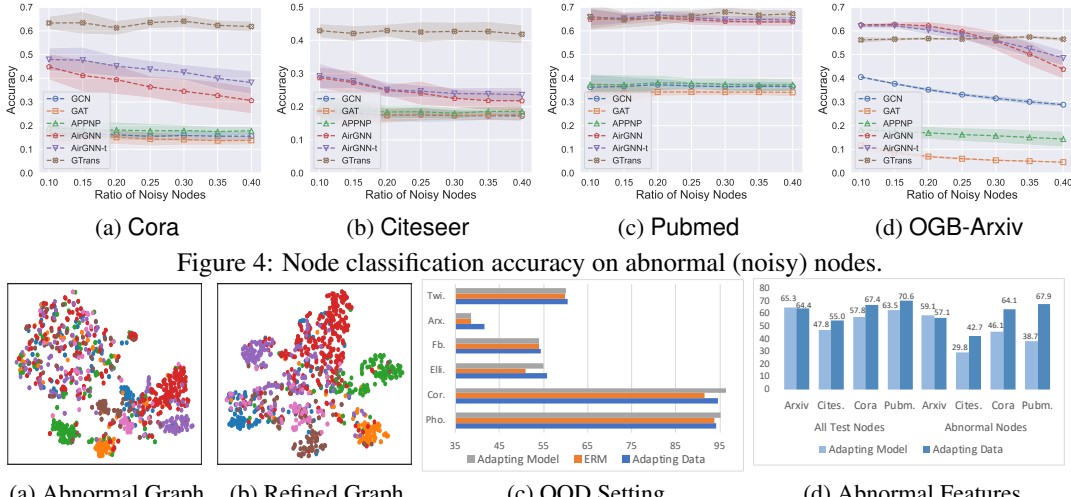

Figure 4: Node classification accuracy on abnormal (noisy) nodes.

(a) Abnormal Graph  (b) Refined Graph  (c) OOD Setting  (d) Abnormal Features

Figure 5: **(a)(b)** T-SNE visualizations of embedding obtained from abnormal graph and transformed graph on Cora. **(c)(d)** Comparison between adapting data and adapting model at test time.

Note that AirGNN-t tunes the message-passing hyper-parameter in AirGNN at test time. For a fair comparison, we tune AirGNN-t based on the performance on both training and validation nodes.

**Results.** For each model, we present the node classification accuracy on both abnormal nodes and all test nodes (i.e., both normal and abnormal ones) in Figure 4 and Figure 8 (See Appendix D.5), respectively. From these figures, we have two observations. First, GTRANS significantly improves GCN in terms of the performance on abnormal nodes and all test nodes for all datasets across all noise ratios. For example, on Cora with 30% noisy nodes, GTRANS improves GCN by 48.2% on abnormal nodes and 31.0% on overall test accuracy. This demonstrates the effectiveness of the graph transformation process in GTRANS in alleviating the effect of abnormal features. Second, GTRANS shows comparable or better performance with AirGNNs, which are the SOTA defense methods for tackling abnormal features. It is worth mentioning that AirGNN-t improves AirGNN by tuning its hyper-parameter at test time, which aligns with our motivation that test-time adaptation can enhance model test performance. To further understand the effect of graph transformation, we provide the visualization of the test node embeddings obtained from abnormal graph (0.3 noise ratio) and transformed graph for Cora in Figures 5a and 5b, respectively. We observe that the transformed graph results in well-clustered node representations, which indicates that GTRANS can promote intra-class compactness and counteract the effect of abnormal patterns.

### 4.3 ROBUSTNESS TO ADVERSARIAL ATTACK

**Setup.** We further evaluate GTRANS under the setting of adversarial attack where we perturb the test graph, i.e., evasion attack. Specifically, we use PR-BCD (Geisler et al., 2021), a scalable attack method, to attack the test graph in OGB-Arxiv. We focus on structural attacks, and vary the perturbation rate, i.e., the ratio of changed edges, from 5% to 25% with a step of 5%. Similar to Section 4.2, we adopt the training classification loss together with the proposed contrastive loss to optimize GTRANS. We use GCN as the backbone and employ four robust baselines implemented by the adversarial attack repository DeepRobust (Li et al., 2020) including GAT (Veličković et al., 2018), RobustGCN (Zhu et al., 2019), SimPGCN (Jin et al., 2021a) and GCNJaccard (Xu et al., 2019) as comparisons. Among them, GCNJaccard pre-processes the attacked graph by removing edges where the similarities of connected nodes are less than a threshold; we tune this threshold at test time based on the performance on both training and validation nodes.

**Results.** Table 3 reports the performances under structural evasion attack. We observe that GTRANS consistently improves the performance of GCN under different perturbation rates of adversarial attack. Particularly, GTRANS improves GCN by a larger margin when the perturbation rate is higher. For example, GTRANS outperforms GCN by over 40% under the 25% perturbation rate. Such observation suggests that GTRANS can counteract the devastating effect of adversarial attacks. In addition, the best performing baseline GCNJaccard also modifies the graph at test time, which demonstrates the importance of test-time graph adaptation. Nonetheless, it consistently underperforms our

Table 3: Node classification accuracy (%) under different perturbation (Ptb.) rates of structure attack.

| Ptb. Rate | GCN | GAT | RobustGCN | SimPGCN | GCNJaccard | GTRANS |
|---|---|---|---|---|---|---|
| 5% | 57.47±0.54 | 64.56±0.43 | 61.55±1.20 | 61.30±0.42 | 65.01±0.26 | **66.29±0.25** |
| 10% | 47.97±0.65 | 61.20±0.70 | 58.15±1.55 | 57.01±0.70 | 63.25±0.30 | **65.16±0.52** |
| 15% | 38.04±1.22 | 58.96±0.59 | 55.91±1.27 | 54.13±0.73 | 61.83±0.29 | **64.40±0.38** |
| 20% | 29.05±0.73 | 57.29±0.49 | 54.39±1.09 | 52.26±0.87 | 60.57±0.34 | **63.44±0.50** |
| 25% | 19.58±2.32 | 55.86±0.53 | 52.76±1.44 | 50.46±0.85 | 59.17±0.39 | **62.95±0.67** |

proposed GTRANS, indicating that a learnable transformation function is needed to achieve better robustness under adversarial attacks, which GCNJaccard does not employ.

**Interpretation.** To understand the modifications made on the graph, we compare several properties among clean graph, attacked graph (20% perturbation rate), graph obtained by GCNJaccard, and graph obtained by GTRANS in Table 13 in Appendix D.6. First, adversarial attack decreases homophily and feature similarity, but GTRANS and GCNJaccard promote such information to alleviate the adversarial patterns. Our experiment also shows that GTRANS removes 77% adversarial edges while removing 30% existing edges from the attacked graph. Second, both GTRANS and GCNJaccard focus on deleting edges from the attacked graph, but GCNJaccard removes a substantially larger amount of edges, which may destroy clean graph structure and lead to sub-optimal performance.

## 4.4 FURTHER ANALYSIS

**Cross-Architecture Transferability.** Since the outcome of GTRANS is a refined graph, it can conceptually be employed by any GNN model. Thus, we can transform the graph based on one pre-trained GNN and test the transformed graph on another pre-trained GNN. To examine such transferability, we perform experiments GCN, APPNP, AirGNN and GAT under the abnormal feature setting with 30% noisy nodes on Cora. The results on all test nodes in Table 4. Note that

Table 4: Transferability.

| Tr\Te | GCN | APPNP | AirGNN | GAT |
|---|---|---|---|---|
| GCN | 67.36 | 70.65 | 70.84 | 58.62 |
| APPNP | 67.87 | 70.39 | 69.59 | 64.46 |
| AirGNN | 68.00 | 70.37 | 72.68 | 64.93 |
| GAT | 54.85 | 60.37 | 65.22 | 54.60 |
| Noisy | 44.29 | 48.26 | 58.51 | 21.23 |

"Tr" stands for GNNs used in TTGT while "Te" denotes GNNs used for obtaining predictions on the transformed graph; "Noisy" indicates the performance on the noisy graph. We observe that the transformed graph yields good performance even outside the scope it was optimized for. We anticipate that such transferability can alleviate the need for costly re-training on new GNNs.

**Adapting Model vs. Adapting Data.** We empirically compare the performance between adapting data and adapting model and consider the OOD and abnormal feature settings. Specifically, we use GCN as the backbone and adapt the model parameters by optimizing the same loss function as used in GTRANS. The results are shown in Figures 5c and 5d. In OOD setting, both adapting model and adapting data can generally improve GCN's performance. Since their performances are still close, it is hard to give a definite answer on which strategy is better. However, we can observe significant performance differences when the graph contains abnormal features: adapting data outperforms adapting model on 3 out of 4 datasets. This suggests that adapting data can be more powerful when the data is perturbed, which aligns with our analysis in Section 3.3.

**Learning Features v.s. Learning Structure.** Since our framework learns both node features and graph structure, we investigate when one component plays a more important role than the other. Our results are shown in Tables 16 and 17 in Appendix D.8. From the tables, we observe that (1) while each component can improve the vanilla performance, feature learning is more crucial for counteracting feature corruption and structure learning is more important for defending structure corruption; and (2) combining them generally yields a better or comparable performance.

## 5 CONCLUSION

GNNs tend to yield unsatisfying performance when the presented data is sub-optimal. To tackle this issue, we seek to enhance GNNs from a data-centric perspective by transforming the graph data at test time. We propose GTRANS which optimizes a contrastive surrogate loss to transform graph structure and node features, and provide theoretical analysis with deeper discussion to understand this framework. Experimental results on distribution shift, abnormal features and adversarial attack have demonstrated the effectiveness of our method. In the future, we plan to explore more applications of our framework such as mitigating degree bias and long-range dependency.

ACKNOLWEDGEMENT

This research is supported by the National Science Foundation (NSF) under grant numbers IIS1845081, IIS1928278, IIS1955285, IIS2212032, IIS2212144, IOS2107215, and IOS2035472, the Army Research Office (ARO) under grant number W911NF-21-1-0198, MSU Foundation, the Home Depot, Cisco Systems Inc, Amazon Faculty Award, Johnson & Johnson and Snap Inc.

ETHICS STATEMENT

To the best of our knowledge, there are no ethical issues with this paper.

REPRODUCIBILITY STATEMENT

To ensure reproducibility of our experiments, we provide our source code at https://github.com/ChandlerBang/GTrans. The hyper-parameters are described in details in the appendix. We also provide a pseudo-code implementation of our framework in the appendix.

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

# A PROOFS

## A.1 THEREOM 1

**Theorem 1.** *Let $\mathcal{L}_c$ denote the classification loss and $\mathcal{L}_s$ denote the surrogate loss, respectively. Let $\rho(G)$ denote the correlation between $\nabla_G \mathcal{L}_c(G, \mathcal{Y})$ and $\nabla_G \mathcal{L}_s(G)$, and let $\epsilon$ denote the learning rate for gradient descent. Assume that $\mathcal{L}_c$ is twice-differentiable and its Hessian matrix satisfies $\|\mathbf{H}(G, \mathcal{Y})\|_2 \leq M$ for all $G$. When $\rho(G) > 0$ and $\epsilon < \frac{2\rho(G)\|\nabla_G \mathcal{L}_c(G, \mathcal{Y})\|_2}{M\|\nabla_G \mathcal{L}_s(G)\|_2}$, we have*

$$\mathcal{L}_c\left(G - \epsilon \nabla_G \mathcal{L}_s\left(G\right), \mathcal{Y}\right) < \mathcal{L}_c\left(G, \mathcal{Y}\right). \tag{8}$$

*Proof.* Given that $\mathcal{L}_c$ is differentiable and twice-differentiable, we perform first-order Taylor expansion with Lagrange form of remainder at $G - \epsilon \nabla_G \mathcal{L}_s(G)$:

$$\mathcal{L}_c\left(G - \epsilon \nabla_G \mathcal{L}_s\left(G\right), \mathcal{Y}\right) \tag{9}$$

$$= \mathcal{L}_c\left(G, \mathcal{Y}\right) - \epsilon \rho\left(G\right) \left\|\nabla_G \mathcal{L}_c\left(G, \mathcal{Y}\right)\right\|_2 \left\|\nabla_G \mathcal{L}_s\left(G\right)\right\|_2 + \frac{\epsilon^2}{2} \nabla_G \mathcal{L}_s(G)^\top \mathbf{H}(G - \epsilon\theta\nabla_G \mathcal{L}_s(G), \mathcal{Y})\nabla_G \mathcal{L}_s(G),$$

where $\theta \in (0, 1)$ is a constant given by Lagrange form of the Taylor's remainder (here we slightly abuse the notation), and $\rho(G)$ is the correlation between $\nabla_G \mathcal{L}_c(G, \mathcal{Y})$ and $\nabla_G \mathcal{L}_s(G)$:

$$\rho\left(G\right) = \frac{\nabla_G \mathcal{L}_c\left(G, \mathcal{Y}\right)^T \nabla_G \mathcal{L}_s\left(G\right)}{\left\|\nabla_G \mathcal{L}_c\left(G, \mathcal{Y}\right)\right\|_2 \left\|\nabla_G \mathcal{L}_s\left(G\right)\right\|_2}. \tag{10}$$

Before we proceed to the next steps, we first show that given a vector $\mathbf{p}$ and a symmetric matrix $\mathbf{A}$, the inequality $\mathbf{p}^\top \mathbf{A}\mathbf{p} \leq \|\mathbf{p}\|_2^2\|\mathbf{A}\|_2$ holds:

$$\begin{aligned}
\mathbf{p}^\top \mathbf{A}\mathbf{p} &= \sum \sigma_i \mathbf{p}^\top \mathbf{u}_i \mathbf{u}_i^\top \mathbf{p} \quad \text{(Performing SVD on } \mathbf{A}\text{, i.e., } \mathbf{A} = \sum \sigma_i \mathbf{u}_i \mathbf{u}_i^\top) \\
&= \sum \sigma_i \mathbf{v}_i^\top \mathbf{v}_i \quad \text{(Let } \mathbf{v} = \mathbf{U}^\top \mathbf{p}, \text{ where } \mathbf{U} = [\mathbf{u}_1; \mathbf{u}_2; \dots; \mathbf{u}_n]) \\
&\leq \sum \sigma_{\max} \mathbf{v}_i^\top \mathbf{v}_i = \sigma_{\max}\|\mathbf{v}\|_2^2 = \sigma_{\max}\|\mathbf{U}^\top \mathbf{p}\|_2^2 \\
&= \sigma_{\max}\|\mathbf{p}\|_2^2 \quad (\mathbf{U} \text{ is an orthogonal matrix}) \\
&= \|\mathbf{p}\|_2^2\|\mathbf{A}\|_2
\end{aligned} \tag{11}$$

Since the Hessian matrix is symmetric, we can use the above inequality to derive:

$$\mathcal{L}_c\left(G - \epsilon \nabla_G \mathcal{L}_s\left(G\right), \mathcal{Y}\right) \tag{12}$$

$$\leq \mathcal{L}_c\left(G, \mathcal{Y}\right) - \epsilon \rho\left(G\right) \left\|\nabla_G \mathcal{L}_c\left(G, \mathcal{Y}\right)\right\|_2 \left\|\nabla_G \mathcal{L}_s\left(G\right)\right\|_2 + \frac{\epsilon^2}{2}\|\nabla_G \mathcal{L}_s(G)\|_2^2\|\mathbf{H}(G - \epsilon\theta\nabla_G \mathcal{L}_s(G), \mathcal{Y})\|_2.$$

Then we rewrite Eq. (12) as:

$$\mathcal{L}_c\left(G - \epsilon \nabla_G \mathcal{L}_s\left(G\right), \mathcal{Y}\right) - \mathcal{L}_c\left(G, \mathcal{Y}\right)$$

$$\leq -\epsilon \rho\left(G\right) \left\|\nabla_G \mathcal{L}_c\left(G, \mathcal{Y}\right)\right\|_2 \left\|\nabla_G \mathcal{L}_s\left(G\right)\right\|_2 + \frac{\epsilon^2}{2}\|\nabla_G \mathcal{L}_s(G)\|_2^2\|\mathbf{H}(G - \epsilon\theta\nabla_G \mathcal{L}_s(G), \mathcal{Y})\|_2. \tag{13}$$

Given $\|\mathbf{H}(G, \mathcal{Y})\|_2 \leq M$, we know

$$\mathcal{L}_c\left(G - \epsilon \nabla_G \mathcal{L}_s\left(G\right), \mathcal{Y}\right) - \mathcal{L}_c\left(G, \mathcal{Y}\right)$$

$$\leq -\epsilon \rho\left(G\right) \left\|\nabla_G \mathcal{L}_c\left(G, \mathcal{Y}\right)\right\|_2 \left\|\nabla_G \mathcal{L}_s\left(G\right)\right\|_2 + \frac{\epsilon^2 M}{2}\|\nabla_G \mathcal{L}_s(G)\|_2^2. \tag{14}$$

By setting $\epsilon = \frac{2\rho(G)\|\nabla_G \mathcal{L}_c(G, \mathcal{Y})\|_2}{M\|\nabla_G \mathcal{L}_s(G)\|_2}$, we have

$$\mathcal{L}_c\left(G - \epsilon \nabla_G \mathcal{L}_s\left(G\right), \mathcal{Y}\right) - \mathcal{L}_c\left(G, \mathcal{Y}\right) = 0. \tag{15}$$

Therefore, when $\epsilon < \frac{2\rho(G)\|\nabla_G \mathcal{L}_c(G, \mathcal{Y})\|_2}{M\|\nabla_G \mathcal{L}_s(G)\|_2}$ and $\rho(G) > 0$, we have

$$\mathcal{L}_c\left(G - \epsilon \nabla_G \mathcal{L}_s\left(G\right), \mathcal{Y}\right) < \mathcal{L}_c\left(G, \mathcal{Y}\right). \tag{16}$$

$\square$

## A.2 THEREOM 2

**Theorem 2.** *Assume that the augmentation function $\mathcal{A}(\cdot)$ generates a data view of the same class for the test nodes and the node classes are balanced. Assume for each class, the mean of the repre-*

*sentations obtained from $Z$ and $\hat{Z}$ are the same. Minimizing the first term in Eq. (6) is approximately minimizing the class-conditional entropy $H(Z|Y)$ between features $Z$ and labels $Y$.*

*Proof.* For convenience, we slightly abuse the notations to replace $\frac{\mathbf{z}_i}{\|\mathbf{z}_i\|}$ and $\frac{\hat{\mathbf{z}}_i}{\|\hat{\mathbf{z}}_i\|}$ with $\mathbf{z}_i$ and $\hat{\mathbf{z}}_i$, respectively. Then we have $\|\mathbf{z}_i\| = \|\hat{\mathbf{z}}_i\| = 1$. Let $Z_k$ denote the set of test samples from class $k$; thus $|Z_k| = \frac{N}{K}$. Let $\stackrel{c}{=}$ denote equality up to a multiplicative and/or additive constant. Then the first term in Eq. (6) can be rewritten as:

$$\sum_{i=1}^{N}(1 - \hat{\mathbf{z}}_i^\top \mathbf{z}_i) = \sum_{i=1}^{N}\left(1 - \hat{\mathbf{z}}_i^\top \mathbf{z}_i\right) \stackrel{c}{=} \sum_{k=1}^{K}\frac{1}{|Z_k|}\sum_{\mathbf{z}_i \in Z_k}\left(-\hat{\mathbf{z}}_i^\top \mathbf{z}_i\right) \tag{17}$$

Let $\mathbf{c}_k$ be the mean of hidden representations from class $k$; then we have $\mathbf{c}_k = \frac{1}{|Z_k|}\sum_{\mathbf{z}_i \in Z_k}\mathbf{z}_i = \frac{1}{|Z_k|}\sum_{\hat{\mathbf{z}}_i \in \hat{Z}_k}\hat{\mathbf{z}}_i$. Now we build the connection between Eq. (17) and $\sum_{i=1}^{K}\sum_{\mathbf{z}_i \in Z_k}\|\mathbf{z}_i - \mathbf{c}_k\|^2$:

$$\sum_{i=1}^{K}\sum_{\mathbf{z}_i \in Z_k}\|\mathbf{z}_i - \mathbf{c}_k\|^2 \tag{18}$$

$$= \sum_{i=1}^{K}\left(\sum_{\mathbf{z}_i \in Z_k}\|\mathbf{z}_i\|^2 - 2\sum_{\mathbf{z}_i \in Z_k}\mathbf{z}_i^\top \mathbf{c}_k + |Z_k|\|\mathbf{c}_k\|^2\right)$$

$$= \sum_{i=1}^{K}\left(\sum_{\mathbf{z}_i \in Z_k}\|\mathbf{z}_i\|^2 - 2\frac{1}{|Z_k|}\sum_{\mathbf{z}_i \in Z_k}\sum_{\hat{\mathbf{z}}_i \in \hat{Z}_k}\hat{\mathbf{z}}_i^\top \mathbf{z}_i + \frac{1}{|Z_k|}\sum_{\mathbf{z}_i \in Z_k}\sum_{\hat{\mathbf{z}}_i \in \hat{Z}_k}\hat{\mathbf{z}}_i^\top \mathbf{z}_i\right)$$

$$= \sum_{i=1}^{K}\left(\sum_{\mathbf{z}_i \in Z_k}\|\mathbf{z}_i\|^2 - \frac{1}{|Z_k|}\sum_{\mathbf{z}_i \in Z_k}\sum_{\hat{\mathbf{z}}_i \in \hat{Z}_k}\hat{\mathbf{z}}_i^\top \mathbf{z}_i\right)$$

$$\stackrel{c}{=} \sum_{i=1}^{K}\left(\frac{1}{|Z_k|}\sum_{\mathbf{z}_i \in Z_k}\sum_{\hat{\mathbf{z}}_i \in \hat{Z}_k}\|\mathbf{z}_i\|^2 - \frac{1}{|Z_k|}\sum_{\mathbf{z}_i \in Z_k}\sum_{\hat{\mathbf{z}}_i \in \hat{Z}_k}\hat{\mathbf{z}}_i^\top \mathbf{z}_i\right)$$

$$= \sum_{i=1}^{K}\left(\frac{1}{|Z_k|}\sum_{\mathbf{z}_i \in Z_k}\sum_{\hat{\mathbf{z}}_i \in \hat{Z}_k}\left(\|\mathbf{z}_i\|^2 - \hat{\mathbf{z}}_i^\top \mathbf{z}_i\right)\right)$$

$$\stackrel{c}{=} \sum_{i=1}^{K}\left(\frac{1}{|Z_k|}\sum_{\mathbf{z}_i \in Z_k}\sum_{\hat{\mathbf{z}}_i \in \hat{Z}_k}\left(-\hat{\mathbf{z}}_i^\top \mathbf{z}_i\right)\right) \tag{19}$$

By comparing Eq. (17) and Eq. (19), the only difference is that Eq. (19) includes more positive pairs for loss calculation. Hence, minimizing Eq. (17) can be viewed as approximately minimizing Eq. (19) or Eq. (18) through sampling positive pairs. As demonstrated in the work (Boudiaf et al., 2020), Eq. (18) can be interpreted as a conditional cross-entropy between $Z$ and another random variable $\bar{Z}$, whose conditional distribution given $Y$ is a standard Gaussian centered around $\mathbf{c}_Y : Z \mid Y \sim \mathcal{N}(\mathbf{c}_Y, \mathbf{I})$:

$$\sum_{\mathbf{z}_i \in Z_k}\|\mathbf{z}_i - \mathbf{c}_k\|^2 = \mathcal{H}(Z \mid Y) + \mathcal{D}_{KL}(Z\|\bar{Z} \mid Y) \geq \mathcal{H}(Z \mid Y) \tag{20}$$

Hence, minimizing the first term in Eq. (6) is approximately minimizing $H(Z|Y)$. $\qquad\square$

**Discussion:** We note that the assumption "the mean of the representations obtained from $Z$ and $\hat{Z}$ are the same" can be inferred by the first assumption about data augmentation. Let $p_k(X)$ denote the distribution of samples with class $k$ and let $x \sim p_k(X)$ denote the sample with class $k$. Recall that we assume the data augmentation function $\mathcal{A}(\cdot)$ is strong enough to generate a data view that can simulate the test data from the same class. In this regard, the new data view can be regarded as an independent sample from the same class, i.e., $\mathcal{A}(x) \sim p_k(X)$. Hence, the expectation of $Z$ and $\hat{Z}$ is the same and we would approximately have that "the mean of $Z$ and $\hat{Z}$ is the same for each class". Particularly, when the number of samples is relatively large, the mean of $Z$ ($\hat{Z}$) would be

close to the true distribution mean. For example, on one graph of Cora, the mean absolute difference between the two mean representations of $Z$ and $\hat{Z}$ are [0.018, 0.009, 0.021, 0.024, 0.016, 0.014, 0.0, 0.016, 0.023] for each class, which are actually very small.

### A.3 A Figurative Example

In Figure 6, we show an example of adversarial attack which causes the aggregated features for two nodes to be the same. Given two nodes $\mathbf{x}_1$ and $\mathbf{x}_2$ and their connections, we are interested in predicting their labels. Assume a mean aggregator is used for aggregating features from the neighbors. Before attack, the aggregated features for them are $\bar{\mathbf{x}}_1 = [0.45]$ and $\bar{\mathbf{x}}_2 = [0.53]$ while after attack the aggregated features become the same $\bar{\mathbf{x}}_1 = \bar{\mathbf{x}}_2 = [0.45]$. In this context, it is impossible to learn a classifier that can distinguish the two nodes.

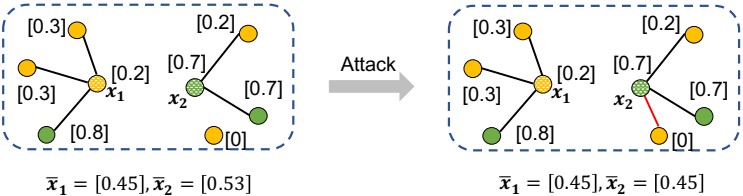

Figure 6: Given two nodes $\mathbf{x}_1$ and $\mathbf{x}_2$ and their connections, we are interested in predicting their labels. The color indicates the node label and "[0.3]" suggests that the associated node feature is 0.3. Assume a mean aggregator is used for aggregating features from the neighbors. **Left:** we show the clean graph without adversarial attack. The aggregated features for the two center nodes are $\bar{\mathbf{x}}_1 = [0.45]$ and $\bar{\mathbf{x}}_2 = [0.53]$. **Right:** we show the attacked graph where the red edge indicates the adversarial edge injected by the attacker. The aggregated features for the two center nodes become $\bar{\mathbf{x}}_1 = [0.45]$ and $\bar{\mathbf{x}}_2 = [0.45]$.

## B Algorithm

We show the detailed algorithm of GTRANS in Algorithm 1. In detail, we first initialize $\Delta_{\mathbf{A}}$ and $\Delta_{\mathbf{X}'}$ as zero matrices and calculate $\mathcal{L}_s$ based on Eq. (6). Since we alternatively optimize $\Delta_{\mathbf{A}}$ and $\Delta_{\mathbf{X}'}$, we update $\Delta_{\mathbf{X}'}$ every $\tau_1$ epochs and update $\Delta_{\mathbf{A}}$ every $\tau_2$ epochs. When the optimization is done, we sample the discrete graph structure for $K$ times and select the one that results in the smallest $\mathcal{L}_s$ as the final adjacency matrix.

## C Datasets and Hyper-Parameters

In this section, we reveal the details of reproducing the results in the experiments. We will release the source code upon acceptance.

### C.1 Out-of-Distribution (OOD) Setting

The out-of-distribution (OOD) problem indicates that the model does not generalize well to the test data due to the distribution gap between training data and test data (Yang et al., 2021a), which is also referred to as distribution shifts. Numerous research studies have been conducted to explore this problem and propose potential solutions (Ganin et al., 2016; Zhu et al., 2021a; Yang et al., 2021b;c; Wu et al., 2022a; Liu et al., 2022a; Chen et al., 2022; Buffelli et al., 2022; Gui et al., 2022; Wu et al., 2022b; You et al., 2023). In the following, we introduce the datasets used for evaluating the methods that tackle the OOD issue in graph domain.

**Dataset Statistics.** For the evaluation on OOD data, we use the datasets provided by Wu et al. (2022a). The dataset statistics are shown in Table 5, which includes three distinct type of distribution shifts: (1) artificial transformation which indicates the node features are replaced by synthetic spurious features; (2) cross-domain transfers which means that graphs in the dataset are from different domains and (3) temporal evolution where the dataset is a dynamic one with evolving nature. Notably, we use the datasets provided by Wu et al. (2022a), which were adopted from the

---

**Algorithm 1:** GTRANS for Test-Time Graph Transformation

---

1  **Input:** Pre-trained model $f_\theta$ and test graph dataset $G_{\text{Te}} = (\mathbf{A}, \mathbf{X}')$.
2  **Output:** Model prediction $\hat{\mathbf{Y}}$ and transformed graph $G'' = (\mathbf{A}', \mathbf{X}'')$.
3  Initialize $\Delta_\mathbf{A}$ and $\Delta_\mathbf{X}$ as zero matrices
4  **for** $t = 0, \ldots, T - 1$ **do**
5      Compute $\mathbf{A}' = \mathbf{A} \oplus \Delta_\mathbf{A}$ and $\mathbf{X}' = \mathbf{X} + \Delta_\mathbf{X}$
6      Compute $\mathcal{L}_s(\Delta_\mathbf{A}, \Delta_\mathbf{X})$ as shown in Eq. (6)
7      **if** $t\%(\tau_1 + \tau_2) < \tau_1$ **then**
8         Update $\Delta_\mathbf{X} \leftarrow \Delta_\mathbf{X} - \eta_1 \nabla_{\Delta_\mathbf{X}} \mathcal{L}_s$
9      **else**
10        Update $\Delta_\mathbf{A} \leftarrow \Pi_\mathcal{P} \left( \Delta_\mathbf{A} - \eta \nabla_{\Delta_\mathbf{A}} \mathcal{L}_s \right)$
11  $\ell_{\text{best}} = \infty$    # store the best loss
12  **for** $k = 0, \ldots, K - 1$ **do**
13      Sample $\mathbf{A}'_0 \sim \text{Bernoulli}(\mathbf{A} \oplus \Delta_\mathbf{A})$
14      Calculate $\mathcal{L}_s$ with $\mathbf{A}'_0$ as input
15      **if** $\mathcal{L}_s < \ell_{best}$ **then**
16         $\ell_{\text{best}} = \mathcal{L}_s$
17         $\mathbf{A}' = \mathbf{A}'_0$
18  $\mathbf{X}' = \mathbf{X} + \Delta_\mathbf{X}$
19  $\hat{\mathbf{Y}} = f_\theta(\mathbf{A}', \mathbf{X}')$
20  **Return:** $\hat{\mathbf{Y}}, (\mathbf{A}', \mathbf{X}')$

---

aforementioned references with manually created distribution shifts. Note that there can be multiple training/validaiton/test graphs. Specifically, Cora and Amazon-Photo have 1/1/8 graphs for training/validation/test sets. Similarly, the splits are 1/1/5 on Twitch-E, 3/2/3 on FB-100, 5/5/33 on Elliptic, and 1/1/3 on OGB-Arxiv.

Table 5: Summary of the experimental datasets that entail diverse distribution shifts.

| Distribution Shift | Dataset | #Nodes | #Edges | #Classes | Train/Val/Test Split | Metric | Adapted From |
|---|---|---|---|---|---|---|---|
| Artificial Transformation | Cora | 2,703 | 5,278 | 10 | Domain-Level | Accuracy | Yang et al. (2016) |
| | Amz-Photo | 7,650 | 119,081 | 10 | Domain-Level | Accuracy | Shchur et al. (2018) |
| Cross-Domain Transfers | Twitch-E | 1,912 9,498 | 31,299 - 153,138 | 2 | Domain-Level | ROC-AUC | Rozemberczki et al. (2021a) |
| | FB100 | 769 41,536 | 16,656 - 1,590,655 | 2 | Domain-Level | Accuracy | Traud et al. (2012) |
| Temporal Evolution | Elliptic | 203,769 | 234,355 | 2 | Time-Aware | F1 Score | Pareja et al. (2020) |
| | OGB-Arxiv | 169,343 | 1,166,243 | 40 | Time-Aware | Accuracy | Hu et al. (2020) |

**Hyper-Parameter Setting.** For the setup of backbone GNNs, we majorly followed Wu et al. (2022a):

(a) GCN: the architecture setup is 5 layers with 32 hidden units for Elliptic and OGB-Arxiv, and 2 layers with 32 hidden units for other datasets, and with batch normalization for all datasets. The learning rate is set to 0.001 for Cora and Amz-Photo, 0.01 for other datasets; the weight decay is set to 0 for Elliptic and OGB-Arxiv, and 0.001 for other datasets.

(b) GraphSAGE: the architecture setup is 5 layers with 32 hidden units for Elliptic and OGB-Arxiv, and 2 layers with 32 hidden units for other datasets, and with batch normalization for all datasets. The learning rate is set to 0.01 for all datasets; the weight decay is set to 0 for Elliptic and OGB-Arxiv, and 0.001 for other datasets.

(c) GAT: the architecture setup is 5 layers for Elliptic and OGB-Arxiv, and 2 layers for other datasets, and with batch normalization for all datasets. Each layer contains 4 attention heads and each head is associated with 32 hidden units. The learning rate is set to 0.01 for all datasets; the weight decay is set to 0 for Elliptic and OGB-Arxiv, and 0.001 for other datasets.

(d) GPR: We use 10 propagation layers and 2 transformation layers with 32 hidden units. The learning rate is set to 0.01 for all datasets; the weight decay is set to 0 for Elliptic and OGB-Arxiv, and 0.001 for other datasets. Note that GPR does not contain batch normalization layers.

For the baseline methods, we tuned their hyper-parameters based on the validation performance. For DropEdge, we search the drop ratio in the range of [0, 0.05, 0.1, 0.15, 0.2, 0.3, 0.5, 0.7]. For Tent, we search the learning rate in the range of [1e-2, 1e-3, 1e-4, 1e-5, 1e-6] and the running epochs in [1, 10, 20, 30]. For EERM, we followed the instruction provided by the original paper. For GTRANS, we alternatively optimize node features for $\tau_1 = 4$ epochs and optimize graph structure $\tau_2 = 1$ epoch. We adopt DropEdge as the augmentation function $\mathcal{A}(\cdot)$ and set the drop ratio to 0.5. We use Adam optimizer for both feature learning and structure learning. We further search the learning rate of feature adaptation $\eta_1$ in [5e-3, 1e-3, 1e-4, 1e-5, 1e-6], learning rate of structure adaptation $\eta_2$ in [0.5, 0.1, 0.01], the modification budget $B$ in [0.5%, 1%, 5%] of the original edges, total epochs $T$ in [5, 10]. We note that the process of tuning hyper-parameters is quick due to the high efficiency of test-time adaptation as we demonstrated in Section 4.1.

**Evaluation Protocol.** For ERM (standard training), we train all the GNN backbones using the common cross entropy loss. For DropEdge, we drop a certain amount of edges at each training epoch. For EERM, it optimizes a bi-level problem to obtain a trained classifier. Note that the aforementioned three methods do not perform any test-time adaptation and their model parameters are fixed during test. For the two test-time adaptation methods, Tent and GTRANS, we first obtain the GNN backbones pre-trained from ERM and adapt the model parameters or graph data at test time, respectively. Furthermore, Tent minimizes the entropy loss while GTRANS minimizes the contrastive surrogate loss.

**Quantifying Distribution Shifts.** Following SR-GNN (Zhu et al., 2021a), we adopt central moment discrepancy (CMD) (Zellinger et al., 2017) as the measurement to quantify the distribution shifts in different graphs. Specifically, given a pre-trained model, we obtain its hidden representation on the training graph and test graphs, denoted as $Z_{\text{tr}}$ and $Z_{\text{te}}$. Then we calculate their distance by the CMD metric, i.e., CMD($Z_{\text{tr}}$), $Z_{\text{te}}$. We show the results in Table 6 and we can observe certain distribution shifts as these values are not small. Let's take the OGB-Arxiv dataset as an example, where we select papers published before 2011 for training, 2011-2014 for validation, and within 2014-2016/2016-2018/2018-2020 for test. In this context, the distribution shift is from the temporal change. In Table 7, we show the CMD values, ERM performance and GTRANS performance. From the table, we can find that (1) the CMD value on the validation graph is essentially smaller than those on test graphs; and (2) GCN performances on test graphs (with larger shifts) are lower than that on the validation graph.

Table 6: CMD values on each individual graph based on the pre-trained GCN.

| GraphID | $G_0$ | $G_1$ | $G_2$ | $G_3$ | $G_4$ | $G_5$ | $G_6$ | $G_7$ | $G_8$ |
|---|---|---|---|---|---|---|---|---|---|
| Amz-Photo | 6.4 | 5.1 | 5.5 | 3.7 | 2.8 | 3.7 | 3.9 | 6.6 | - |
| Cora | 5.4 | 4.2 | 4.8 | 6.3 | 5.5 | 4.8 | 4.6 | 5.4 | - |
| Elliptic | 80.2 | 90.8 | 114.3 | 86.5 | 789.3 | 781.6 | 99.4 | 100.4 | 150.6 |
| OGB-Arxiv | 14.7 | 20.6 | 10.4 | - | - | - | - | - | - |
| FB-100 | 29.7 | 16.9 | 32.9 | - | - | - | - | - | - |
| Twitch-E | 8.6 | 6.1 | 9.0 | 8.4 | 9.7 | - | - | - | - |

Table 7: CMD values and the performances of ERM and GTRANS on OGB-Arxiv

| Method | 2011-2014 (Val) | 2014-2016 | 2014-2016 | 2016-2018 | 2018-2020 |
|---|---|---|---|---|---|
| CMD | 2.5 | 14.7 | 14.7 | 20.6 | 10.4 |
| ERM | 45.32±0.50 | 41.29±1.13 | 41.29±1.13 | 38.69±1.33 | 35.78±1.81 |
| GTRANS | 45.82±0.38 | 44.03±0.95 | 44.03±0.95 | 41.90±1.28 | 38.81±1.47 |

## C.2   ABNORMAL FEATURES

**Dataset Statistics.** In these two settings, we choose the original version of popular benchmark datasets Cora, Citeseer, Pubmed and OGB-Arxiv. The statistics for these datasets are shown in Table 8. Note that we only have one test graph, and the injection of abnormal features or adversarial attack happens after the training process of backbone model, which can be viewed as evasion attack.

Table 8: Dataset statistics for experiments on abnormal features and adversarial attack.

| Dataset | Classes | Nodes | Edges | Features | Training Nodes | Validation Nodes | Test Nodes |
|---|---|---|---|---|---|---|---|
| Cora | 7 | 2708 | 5278 | 1433 | 20 per class | 500 | 1000 |
| Citeseer | 6 | 3327 | 4552 | 3703 | 20 per class | 500 | 1000 |
| Pubmed | 3 | 19717 | 44324 | 500 | 20 per class | 500 | 1000 |
| OGB-Arxiv | 40 | 169343 | 1166243 | 128 | 54% | 18% | 28% |

**Hyper-Parameter Settings.** We closely followed AirGNN (Liu et al., 2021a) to set up the hyper-parameters for the baselines:

(a) GCN: the architecture setup is 2 layers with 64 hidden units without batch normalization for Cora, Citeseer and Pubmed, and 3 layers with 256 hidden units with batch normalization for OGB-Arxiv. The learning rate is set to 0.01.

(b) GAT: the architecture setup is 2 layers with 8 hidden units in each of the 8 heads without batch normalization for Cora, Citeseer and Pubmed, and 3 layers with 32 hidden units in each of the 8 heads with batch normalization for OGB-Arxiv. The learning rate is set to 0.005.

(c) APPNP: the architecture setup is 2-layer transformation with 64 hidden units and 10-layer propagation without batch normalization for Cora, Citeseer and Pubmed; the architecture is set to 3-layer transformation with 256 hidden units and 10-layer propagation with batch normalization for OGB-Arxiv. The learning rate is set to 0.01.

(d) AirGNN: The architecture setup is the same as APPNP and the hyper-parameter $\lambda$ is set to 0.3 for OGB-Arxiv and 0.5 for other datasets.

(e) AirGNN-t: The architecture setup is the same as AirGNN but we tune the hyper-parameter $\lambda$ in AirGNN based on performance on the combination of training and validation nodes at test stage. This is because the test graph has the same graph structure as the training graph; thus we can take advantage of the label information of training nodes (as well as validation nodes) to tune the hyper-parameters. Specifically, we search $\lambda$ in the range of [0, 0.1, 0.2, 0.3, 0.4, 0.5, 0.6, 0.7, 0.8, 0.9] for each noise ratio.

For the setup of GTRANS, we alternatively optimize node features for $\tau_1 = 4$ epochs and optimize graph structure $\tau_2 = 1$ epoch. We adopt DropEdge as the augmentation function $\mathcal{A}(\cdot)$ and set the drop ratio to 0.5. We use Adam optimizer for both feature learning and structure learning. We further search the learning rate of feature adaptation $\eta_1$ in [1, 1e-1, 1e-2], total epochs $T$ in [10, 20]. The modification budget $B$ to 5% of the original edges and the learning rate of structure adaptation $\eta_2$ is set to 0.1. It is worth noting that *we use a weighted combination of contrastive loss and training classification loss*, i.e., $\mathcal{L}_{\text{train}} + \lambda \mathcal{L}_s$, instead of optimizing the contrastive loss alone. We adaopt this strategy because that the training graph and the test graph were the same graph before the injection of abnormal features. Here the $\lambda$ is tuned in the range of [1e-2, 1e-3, 1e-4]. We study the effects of contrastive loss and training classification loss in Appendix D.7.

## C.3 ADVERSARIAL ATTACK

**Dataset Statistics.** We used OGB-Arxiv for the adversarial attack experiments and the dataset statistics can be found in Table 8. Again, we only have one test graph for this dataset.

**Hyper-Parameter Settings.** The setup of GCN and GAT is the same as that in the setting of abnormal features. For the defense methods including SimPGCN, RobustGCN and GCNJaccard, we use the DeepRobust (Li et al., 2020) library to implement them. For GCNJaccard, we tune its threshold hyper-parameter in the range of [0.1, 0.2, 0.3, 0.4, 0.5, 0.6, 0.7, 0.8, 0.9]. The hyper-parameter is also tuned based on the performance of training and validation nodes (same as Appendix C.2). Note that the popular defenses ProGNN (Jin et al., 2020) and GCNSVD (Entezari et al., 2020) were not included because they throw OOM error due to the expensive eigen-decomposition operation.

We use the official implementation of the scalable attack PR-BCD (Geisler et al., 2021) to attack the test graph. We note that when performing adversarial attacks, the setting is more like transductive setting where the training graph and test graph are the same. However, the test graph becomes different from the training graph after the attack. Since the training graph and test graph were originally the same graph, *we use a weighted combination of contrastive loss and training classification loss,*

i.e., $\mathcal{L}_{\text{train}} + \lambda\mathcal{L}_s$, instead of optimizing the contrastive loss alone. For the setup of GTRANS, we alternatively optimize node features for $\tau_1 = 1$ epoch and optimize graph structure $\tau_2 = 4$ epochs. We fix the learning rate of feature adaptation $\eta_1$ to 1e-3, learning rate of structure adaptation $\eta_2$ to 0.1, $\lambda$ to 1, total epochs $T$ to 50 and modification budget $B$ to 30% of the original edges.

## C.4 HARDWARE AND SOFTWARE CONFIGURATIONS.

We perform experiments on NVIDIA Tesla V100 GPUs. The GPU memory and running time reported in Table 2 are measured on one single V100 GPU. Additionally, we use eight CPUs, with the model name as Intel(R) Xeon(R) Platinum 8260 CPU @ 2.40GHz. The operating system we use is CentOS Linux 7 (Core).

## D MORE EXPERIMENTAL RESULTS

### D.1 COMPARISON TO GRAPH DOMAIN ADAPTATION

Our work is related to graph domain adaptation (GraphDA) (Shen et al., 2020; Wu et al., 2020; Zhu et al., 2021a), but they are also **highly different**. In Table 9, we summarize the differences between GraphDA and GTRANS. In detail, there are the following differences:

(a) **Data and losses**: GraphDA methods optimize the loss function based on both labeled source data (training data) and unlabeled target data (test data), while GTRANS only requires target data during inference. Hence, GraphDA methods are infeasible when access to the source data is prohibited such as online service.
(b) **Parameter**: To our best knowledge, existing GraphDA methods are model-centric approaches while GTRANS is a data-centric approach. GTRANS adapts the data instead of the model, which can be more useful in some settings as we showed in the Example of Section 3.3.
(c) **Efficiency**: GraphDA is indeed a training-time adaptation and for each given test graph, it would require training the model on the source and target data. Thus, it is much less efficient than GTRANS, especially when we have multiple test graphs (e.g., 33 test graphs for Elliptic).

Table 9: Comparison between GraphDA and GTRANS. They differ by their data and losses.

| Setting | Source | Target | Train Loss | Test Loss | Parameter | Efficiency |
|---------|--------|--------|------------|-----------|-----------|------------|
| GraphDA | $G_{\text{tr}}$ | $G_{\text{te}}$ | $\mathcal{L}(G_{\text{tr}}, \mathcal{Y}_{\text{tr}}) + \mathcal{L}(G_{\text{tr}}, G_{\text{te}})$ | - | $f_\theta$ | Low |
| GTRANS | - | $G_{\text{te}}$ | - | $\mathcal{L}(G_{\text{te}})$ | $G_{\text{te}}$ | High |

To compare their empirical performance, we include two GraphDA methods (SR-GNN (Zhu et al., 2021a) and UDA-GCN (Wu et al., 2020)) and one general domain adaptation method (DANN (Ganin et al., 2016)). SR-GNN regularizes the model's performance on the source and target domains. Note that SR-GNN is originally developed under the transductive setting where the training graph and test graph are the same. To apply SR-GNN in our OOD setting, we assume the test graph is available during the training stage of SR-GNN, as typically done in domain adaptation methods. UDA-GCN is another work that tackles graph data domain adaptation, which exploits local and global information for different domains. In addition, we also include DANN, which adopts an adversarial domain classifier to promote the similarity of feature distributions between different domains. We followed the authors' suggestions in their paper to tune the hyper-parameters and the results are shown in Table 10. On the one hand, we can observe that GraphDA methods generally improve the performance of GCN under distribution shift and SRGNN is the best performing baseline. On the other hand, GTRANS performs the best on all datasets except Amz-Photo. On Amz-Photo, GTRANS does not improve as much as SR-GNN, which indicates that joint optimization over source and target is necessary for this dataset. However, recall that domain adaptation methods are less efficient due to the joint optimization on source and target: the adaptation time of SR-GNN on the 8 graphs of Amz-Photo is 83.5s while that of GTRANS is 4.9s (plus pre-training time 10.1s). Overall, test-time graph transformation exhibits strong advantages of effectiveness and efficiency.

Table 10: Performance comparison between GTRANS and graph domain adaptation methods.

| Method | Amz-Photo | Cora | Elliptic | FB-100 | OGB-Arxiv | Twitch-E |
|---|---|---|---|---|---|---|
| ERM | 93.79±0.97 | 91.59±1.44 | 50.90±1.51 | 54.04±0.94 | 38.59±1.35 | 59.89±0.50 |
| UDA-GCN | 91.70±0.35 | 92.65±0.46 | 51.57±1.31 | 54.11±0.54 | 39.43±0.71 | 52.12±0.38 |
| DANN | 94.08±0.21 | 92.89±0.64 | 53.00±0.97 | 51.53±1.47 | 36.60±1.26 | 60.13±0.53 |
| SRGNN | **94.64±0.17** | 94.08±0.28 | 51.94±0.81 | 54.08±1.10 | 38.92±0.65 | 59.21±0.51 |
| GTRANS | 94.13±0.77 | **94.66±0.63** | **55.88±3.10** | **54.32±0.60** | **41.59±1.20** | **60.42±0.86** |

## D.2 COMPARISON TO GRAPH STRUCTURE LEARNING

Our work is also relevant to graph structure learning (GSL) (Franceschi et al., 2019; Jin et al., 2020; Chen et al., 2020; Zhao et al., 2021b; Rozemberczki et al., 2021b; Halcrow et al., 2020; Fatemi et al., 2021) which learns the graph structure during the training time while not adapting the graph structure at test stage. Our proposed test-time graph transform is essentially different from these works as we do not modify the training data but only the test data. It can be of interest to adopt GSL method at test time by also adapting the test graph structure. However, most existing GSL methods optimize the cross entropy loss defined on the labels to update graph structure, thus not applicable in the absence of test labels. One exception is SLAPS (Fatemi et al., 2021) which utilizes a self-supervised loss together with the cross entropy loss to optimize the graph structure. However, the default setting in SLAPS is generating structure for raw data points (with no given graph structure). Hence, using SLAPS for our settings requires considerable changes. Furthermore, we highlight two more weaknesses of SLAPS compared to GTRANS.

(a) **Introducing additional parameters**. SLAPS uses a denoising loss as self-supervision. In detail, it first injects noise into node features and trains a denoising autoencoder to denoise the noisy features. This introduces additional parameters from the denoising autoencoder and inevitably changes the model architecture.

(b) **Not learning features**. As other GSL methods, SLAPS does not learn node features. We argue that feature learning is highly important under the abnormal feature setting as shown in Table 16. For example, structure learning only improves GCN by 2% on OGB-Arxiv while feature learning can improve GCN by 20%. Thus, without the feature learning component, the performance will significantly drop when encountering noisy features.

Since GTRANS is highly versatile and we can use any self-supervised loss as the surrogate loss, we can simply replace the contrastive loss in Eq. (6) with the denoising loss of SLAPS instead of paying considerable efforts in adjusting SLAPS. We refer to the loss used for denoising as SLAPS loss and adopt it for TTGT. Note that we first train the parameters of the DAE used for denoising while keeping the pre-trained model fixed. Then we fix both DAE and the pre-trained model and optimize the test graph for TTGT. The results are shown in Table 11. From the table, we can observe that the SLAPS loss (or feature denoising loss) does not work as well as the contrastive loss.

Table 11: Comparison between SLAPS loss and our contrastive loss.

| | Amz-Photo | Cora | Elliptic | FB-100 | OGB-Arxiv | Twitch-E |
|---|---|---|---|---|---|---|
| None | 93.79±0.97 | 91.59±1.44 | 50.90±1.51 | 54.04±0.94 | 38.59±1.35 | 59.89±0.50 |
| SLAPS | 93.97±1.04 | 91.41±1.23 | 50.54±1.81 | 54.08±0.76 | 41.38±1.35 | 59.85±0.68 |
| $\mathcal{L}_s$ in Eq. (6) | **94.13±0.77** | **94.66±0.63** | **55.88±3.10** | **54.32±0.60** | **41.59±1.20** | **60.42±0.86** |

## D.3 COMPARISON TO AD-GCL

Next, we compare our method with a graph contrastive learning method with learnable augmentation AD-GCL (Suresh et al., 2021). Since AD-GCL is originally designed for graph classification as a pre-training strategy, the direct empirical comparison between AD-GCL and GTRANS is not easy. However, due to the flexibility of GTRANS, we can integrate AD-GCL into our TTGT framework, denoted as TTGT+AD-GCL. We present the empirical results in Table 12. We can observe that

TTGT+AD-GCL generally performs worse than GTRANS except on Amz-Photo, which indicates that GTRANS is a stronger realization of TTGT. Furthermore, we highlight some key differences between it and GTRANS.

(a) AD-GCL requires optimization of a min-max problem which involves parameters of graph structure and model. Thus, adopting it for TTGT would change the pre-trained model architecture.

(b) AD-GCL only augments the graph structure while not learning the features. We argue that feature learning is highly important under the abnormal feature setting as shown in Table 16. For example, structure learning only improves GCN by 2% on OGB-Arxiv while feature learning can improve GCN by 20%. Thus, without the feature learning component, the performance will significantly drop when encountering noisy features.

(c) According to Eq. (9) in the AD-GCL paper, it calculates the similarities between all samples within each mini-batch. When we increase the batch size, we would easily get the out-of-memory issue while a small mini-batch will slow down the learning process. As a consequence, TTGT+AD-GCL is less efficient than GTRANS: the adaptation time of TTGT+AD-GCL on OGB-Arxiv is 12.7s while that of GTRANS is 2.6s.

Table 12: Comparison between GTRANS and AD-GCL under the TTGT framework.

|  | Amz-Photo | Cora | Elliptic | FB-100 | OGB-Arxiv | Twitch-E |
|---|---|---|---|---|---|---|
| ERM | 93.79±0.97 | 91.59±1.44 | 50.90±1.51 | 54.04±0.94 | 38.59±1.35 | 59.89±0.50 |
| TTGT+AD-GCL | **94.96±0.52** | 92.38±1.35 | 54.38±2.77 | 53.81±0.87 | 39.16±0.98 | 59.78±0.65 |
| GTRANS | 94.13±0.77 | **94.66±0.63** | **55.88±3.10** | **54.32±0.60** | **41.59±1.20** | **60.42±0.86** |

## D.4 OUT-OF-DISTRIBUTION (OOD) SETTING

To show the performance on individual test graphs, we choose GCN as the backbone model and include the box plot on all test graphs within each dataset in Figure 7. We observe that GTRANS generally improves over each test graph within each dataset, which validates the effectiveness of test-time graph transformation.

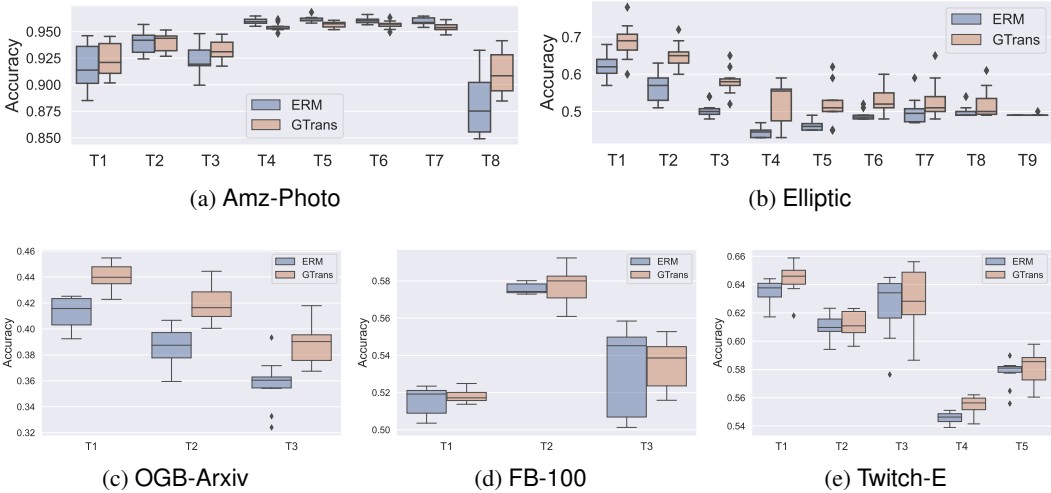

Figure 7: Classification performance on individual test graphs within each dataset for OOD setting.

## D.5 ABNORMAL FEATURES

For each model, we present the node classification accuracy on all test nodes (i.e., both normal and abnormal ones) in Figure 8. GTRANS significantly improves GCN in terms of the performance on all test nodes for all datasets across all noise ratios. For example, on Cora with 30% noisy nodes,

GTRANS improves GCN by 31.0% on overall test accuracy. These results further validate that the proposed GTRANS can produce expressive and generalizable representations.

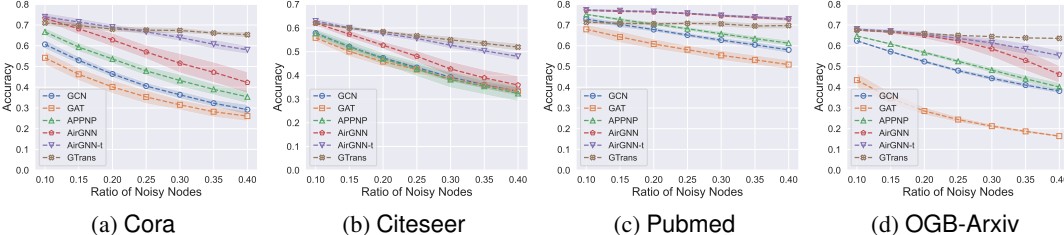

(a) Cora        (b) Citeseer        (c) Pubmed        (d) OGB-Arxiv

Figure 8: Overall node classification accuracy under the setting of abnormal features. GTRANS significantly improves the performance of GCN on both abnormal nodes and overall nodes.

## D.6 INTERPRETATION ON THE REFINED GRAPH FOR ADVERSARIAL ATTACK SETTING

To understand the modifications made on the graph, we compare several properties among clean graph, attacked graph (20% perturbation rate), graph obtained by GCNJaccard, and graph obtained by GTRANS in Table 13. We follow the definition in (Zhu et al., 2020) to measure homophily; "Pairwise Feature Similarity" is the averaged feature similarity among all pairs of connected nodes; "#Edge+/-" indicates the number of edges that the modified graph adds/deletes compared to the clean graph. From Table 13, we observe that first, adversarial attack decreases homophily and feature similarity, but GTRANS and GCNJaccard promote such information to defend against adversarial patterns. Second, both GTRANS and GCNJaccard focus on deleting edges from the attacked graph, but GCNJaccard removes a substantially larger amount of edges, which may destroy clean graph structure and lead to suboptimal performance.

Table 13: Statistics of modified graphs. GTRANS promotes homophily and feature similarity.

|  | GTRANS | GCNJaccard | Attacked | Clean |
|---|---|---|---|---|
| Homophily | 0.689 | 0.636 | 0.548 | 0.654 |
| Pairwise Feature Similarity | 0.825 | 0.863 | 0.809 | 0.827 |
| #Edges | 1,945k | 1,754k | 2,778k | 2,316k |
| #Edge+ | 108k | 118k | 463k | - |
| #Edge- | 479k | 679k | 0.6k | - |

## D.7 ABLATION STUDY ON SURROGATE LOSS

Since we optimized a combined loss in the settings of abnormal features and adversarial attack, we now perform ablation study to examine the effect of each component. We choose GCN as the backbone model and choose 0.3 noise ratio for abnormal features. The results for abnormal features and adversarial attack are shown in Tables 14 and 15, respectively. "None" indicates the vanilla GCN without any test-time adaptation and "Combined" indicates jointly optimizing a combination of the two losses. From the two tables, we can conclude that (1) both $\mathcal{L}_s$ and $\mathcal{L}_{\text{train}}$ help counteract abnormal features and adversarial attack; and (2) optimizing the combined loss generally outperforms optimizing $\mathcal{L}_s$ or $\mathcal{L}_{\text{train}}$ alone.

## D.8 ABLATION STUDY ON FEATURE LEARNING AND STRUCTURE LEARNING

In this subsection, we investigate the effects of the feature learning component and structure learning component. We show results for abnormal features and adversarial attack in Tables 16 and 17, respectively. Note that "None" indicates the vanilla GCN without any test-time adaptation; "$\mathbf{A}'$" or "$\mathbf{X}'$" is the variants of GTRANS which solely learns structure or node features; "Both" indicates the method GTRANS that learn both structure and node features. From Table 16, we observe that (1) while both feature learning and structure learning can improve the vanilla performance, feature

Table 14: Performance comparison when optimizing different losses for abnormal feature setting. Both $\mathcal{L}_s$ and $\mathcal{L}_{\text{train}}$ help counteract abnormal features; optimizing the combined loss generally outperforms optimizing $\mathcal{L}_s$ or $\mathcal{L}_{\text{train}}$ alone.

| Dataset | All Test Nodes | | | | Abnormal Nodes | | | |
|---|---|---|---|---|---|---|---|---|
| | None | $\mathcal{L}_s$ | $\mathcal{L}_{\text{train}}$ | Combined | None | $\mathcal{L}_s$ | $\mathcal{L}_{\text{train}}$ | Combined |
| OGB-Arxiv | 44.29±1.20 | 46.70±1.20 | 64.60±0.22 | 64.64±0.24 | 31.50±1.12 | 35.22±1.17 | 57.54±0.93 | 57.69±0.93 |
| Citeseer | 39.26±2.02 | 45.41±2.71 | 54.97±1.55 | 52.54±1.08 | 17.30±1.86 | 32.93±2.81 | 42.67±2.78 | 44.10±2.97 |
| Cora | 36.35±1.87 | 48.71±3.02 | 66.77±2.54 | 67.29±1.44 | 15.80±2.33 | 35.40±4.05 | 61.67±3.64 | 63.90±2.55 |
| Pubmed | 62.72±1.20 | 65.49±1.65 | 66.56±0.64 | 70.55±1.55 | 36.47±1.85 | 56.77±3.60 | 60.20±1.97 | 67.93±2.11 |

Table 15: Performance comparison when optimizing different losses for adversarial attack setting. Both $\mathcal{L}_s$ and $\mathcal{L}_{\text{train}}$ help counteract adversarial attack; optimizing the combined loss generally outperforms optimizing $\mathcal{L}_s$ or $\mathcal{L}_{\text{train}}$ alone. $r$ denotes the perturbation rate.

| Loss | $r$=5% | $r$=10% | $r$=15% | $r$=20% | $r$=25% |
|---|---|---|---|---|---|
| None | 57.47±0.54 | 47.97±0.65 | 38.04±1.22 | 29.05±0.73 | 19.58±2.32 |
| $\mathcal{L}_s$ | 62.40±0.45 | 59.76±0.93 | 57.85±1.03 | 55.26±1.35 | 52.64±2.35 |
| $\mathcal{L}_{\text{train}}$ | 65.54±0.25 | 64.00±0.31 | 62.99±0.34 | 61.95±0.40 | 61.55±0.58 |
| Combined | 66.29±0.25 | 65.16±0.52 | 64.40±0.38 | 63.44±0.50 | 62.95±0.67 |

learning is more powerful than structure learning; (2) combining them does not seem to further improve the performance but it achieves a comparable performance to sole feature learning. From Table 17, we observe that (1) while both feature learning and structure learning can improve the vanilla performance, structure learning is more powerful than feature learning; and (2) combining them can further improve the performance. From these observations, we conclude that (1) feature learning is more crucial for counteracting feature corruption and structure learning is more important for defending structure corruption; and (2) combining them always yields a better or comparable performance.

Table 16: Ablation study on feature learning and structure learning for abnormal feature setting. While both feature learning and structure learning can improve the vanilla performance, feature learning is more powerful than structure learning. Combining them does not seem to further improve the performance but it achieves a comparable performance to sole feature learning.

| Dataset | All Test Nodes | | | | Abnormal Nodes | | | |
|---|---|---|---|---|---|---|---|---|
| | None | $\mathbf{A}'$ | $\mathbf{X}'$ | Both | None | $\mathbf{A}'$ | $\mathbf{X}'$ | Both |
| OGB-Arxiv | 44.29±1.20 | 46.02±1.09 | 64.88±0.23 | 64.64±0.24 | 31.50±1.12 | 31.96±1.05 | 58.12±0.83 | 57.69±0.93 |
| Citeseer | 39.26±2.02 | 39.67±1.96 | 54.99±1.55 | 54.97±1.55 | 17.30±1.86 | 17.13±1.81 | 42.73±2.81 | 42.67±2.78 |
| Cora | 36.35±1.87 | 37.02±1.82 | 67.40±1.62 | 67.29±1.44 | 15.80±2.33 | 15.67±2.15 | 64.17±3.18 | 63.90±2.55 |
| Pubmed | 62.72±1.20 | 62.50±1.21 | 70.53±1.52 | 70.55±1.55 | 36.47±1.85 | 36.57±1.96 | 67.90±2.07 | 67.93±2.11 |

## D.9 COMPARING DIFFERENT SELF-SUPERVISED SIGNALS

As there can be other choices to guide our test-time graph transformation process, we examine the effects of other self-supervised signals. We choose the OOD setting to perform experiments and consider the following two parameter-free self-supervised loss:

(a) **Reconstruction Loss.** Data reconstruction is considered as a good self-supervised signal and we can adopt link reconstruction (Kipf & Welling, 2016b) as the guidance. Minimizing the reconstruction loss is equivalent to maximizing the similarity for connected nodes, which encourages the connected nodes to have similar representations.

(b) **Entropy Loss.** Entropy loss calculates the entropy of the model prediction. Minimizing the entropy can force the model to be certain about the prediction. It has been demonstrated effective in Tent (Wang et al., 2021) when adapting batch normalization parameters.

Table 17: Ablation study on feature learning and structure learning for adversarial structural attack setting. While both feature learning and structure learning can improve the vanilla performance, structure learning is more powerful than feature learning. Combining them can further improve the performance.

| Param | $r=5\%$ | $r=10\%$ | $r=15\%$ | $r=20\%$ | $r=25\%$ |
|---|---|---|---|---|---|
| None | 57.47±0.54 | 47.97±0.65 | 38.04±1.22 | 29.05±0.73 | 19.58±2.32 |
| $\mathbf{X}'$ | 64.16±0.24 | 61.59±0.29 | 60.07±0.32 | 59.04±0.49 | 58.82±0.68 |
| $\mathbf{A}'$ | 65.93±0.32 | 64.31±0.71 | 63.14±0.39 | 61.42±0.58 | 60.18±1.53 |
| Both | 66.29±0.25 | 65.16±0.52 | 64.40±0.38 | 63.44±0.50 | 62.95±0.67 |

(c) **SLAPS Loss.** SLAPS (Fatemi et al., 2021) utilizes self-supervision to guide the graph structure learning process. Specifically, it injects random noise into node features and employs a denoising autoencoder (DAE) to denoise the node features. We refer to the loss used for denoising as SLAPS loss and adopt it for TTGT. Note that we first train the parameters of the DAE used for denoising while keeping the pre-trained model fixed. Then we fix both DAE and the pre-trained model and optimize the test graph for TTGT.

We summarize the results in Table 18. From the table, we observe that in most of the cases, the above three losses underperform our proposed surrogate loss and even degrade the vanilla performance. It validates the effectiveness of our contrastive loss in guiding the test-time graph transformation.

Table 18: Comparison of different self-supervised signals for OOD setting. The reconstruction loss and entropy loss generally underperform our proposed loss.

| | Amz-Photo | Cora | Elliptic | FB-100 | OGB-Arxiv | Twitch-E |
|---|---|---|---|---|---|---|
| None | 93.79±0.97 | 91.59±1.44 | 50.90±1.51 | 54.04±0.94 | 38.59±1.35 | 59.89±0.50 |
| Recon | 93.77±1.01 | 91.37±1.41 | 49.33±1.37 | 53.94±1.03 | **44.93±4.06** | 59.17±0.77 |
| Entropy | 93.67±0.98 | 91.54±1.14 | 49.93±1.56 | 54.29±0.97 | 41.11±2.19 | 59.48±0.64 |
| SLAPS | 93.97±1.04 | 91.41±1.23 | 50.54±1.81 | 54.08±0.76 | 41.38±1.35 | 59.85±0.68 |
| $\mathcal{L}_s$ in Eq. (6) | **94.13±0.77** | **94.66±0.63** | **55.88±3.10** | **54.32±0.60** | 41.59±1.20 | **60.42±0.86** |

**Gradient Correlation.** In Figure 2, we have empirically verified the effectiveness of Theorem 1 when adopting the surrogate loss in Eq. (6) as $\mathcal{L}_s$. We further plot the values of $\rho(G)$ with different surrogate losses (i.e., entropy, reconstruction and SLAPS) and $\mathcal{L}_c$ on one test graph in Cora in Figure 9. We can observe that a positive $\rho(G)$ generally reduces the test classification loss. For example, when using entropy loss, the test loss generally reduces when $\rho(G)$ is positive and starts to increase after $\rho(G)$ becomes negative.

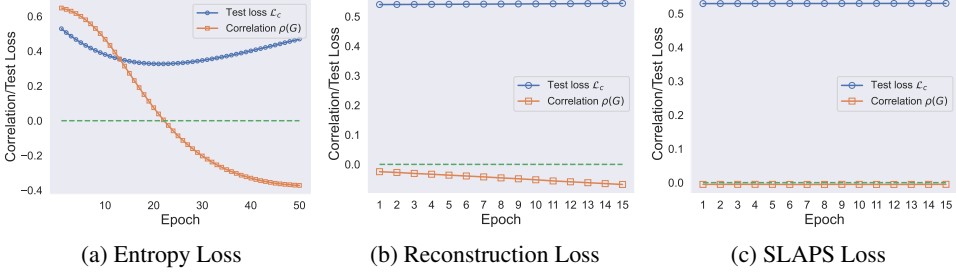

(a) Entropy Loss      (b) Reconstruction Loss      (c) SLAPS Loss

Figure 9: The relationship between $\rho(G)$ and $\mathcal{L}_c$ when adopting different surrogate losses.

### D.10 SENSITIVITY TO HYPER-PARAMETER $B$

In this subsection, we examine the sensitivity of GTRANS' performance with respect to the perturbation budget, i.e., hyper-parameter $B$. Specifically, we vary the value of $B$ in the range of

$\{0.5\%, 1\%, 5\%, 10\%, 20\%, 30\%\}$ and perform experiments on the OGB-Arxiv dataset for the three settings in Table 19. Specifically, "Abn. Feat" stands for abnormal feature setting with 30% noisy feature while "Adv. Attack" stands for the adversarial attack setting with 20% perturbation rate. From the table, we can observe budget $B$ has a smaller effect on OOD and abnormal feature settings while highly impacting the performance under structural adversarial attack. This is because most of the changes made by adversarial attack are edge injections as shown in Table 13, and we need to use a large budget $B$ to remove adversarial patterns. By contrast, GTRANS is much less sensitive to the value of $B$ in the other two settings.

Table 19: The change of model performance when varying budget $B$ on OGB-Arxiv.

| Setting | $B$=0.5% | $B$=1% | $B$=5% | $B$=10% | $B$=20% | $B$=30% |
|---|---|---|---|---|---|---|
| OOD | 40.52 | 40.69 | 41.32 | 41.40 | 41.70 | 41.65 |
| Abn. Feat. | 64.78 | 64.80 | 64.64 | 64.60 | 64.57 | 64.57 |
| Adv. Attack | 56.66 | 56.89 | 58.30 | 59.93 | 62.31 | 63.47 |

### D.11 DIFFERENT AUGMENTATIONS USED IN CONTRASTIVE LOSS

In Eq. (6), we used DropEdge as the augmentation function $\mathcal{A}(\cdot)$ to obtain the augmented view. In practice, the choice of augmentation can be flexible and here we explore two other choices: node dropping (You et al., 2020) and subgraph sampling (Zhu et al., 2021b). We perform experiments on OOD setting with GCN as the backbone model and report the results in Table 20. Specifically, we adopt a ratio of 0.05 for node dropping, and ratios of 0.05 and 0.5 for DropEdge. From the table, we can observe that (1) GTRANS with any of the three augmentations can greatly improve the performance of GCN under distribution shift, and (2) different augmentations lead to slightly different performance on different datasets.

Table 20: Performance of GTRANS with different augmentation used in contrastive loss.

| Augmentation | Amz-Photo | Cora | Elliptic | FB-100 | OGB-Arxiv | Twitch-E |
|---|---|---|---|---|---|---|
| Node Dropping | **94.45±0.70** | 95.00±0.65 | 56.57±2.99 | 54.15±0.60 | 39.95±1.11 | 60.38±0.74 |
| Subgraph Sampling | 94.18±0.75 | 94.95±0.64 | 55.40±3.00 | **54.51±0.56** | 41.44±1.17 | **60.52±0.80** |
| DropEdge (0.05) | 94.43±0.68 | **95.10±0.66** | **56.78±2.86** | 54.17±0.60 | 40.19±1.08 | 60.31±0.74 |
| DropEdge (0.5) | 94.13±0.77 | 94.66±0.63 | 55.88±3.10 | 54.32±0.60 | **41.59±1.20** | 60.42±0.86 |
| ERM | 93.79±0.97 | 91.59±1.44 | 50.90±1.51 | 54.04±0.94 | 38.59±1.35 | 59.89±0.50 |

