# OpenReview forum: "Empowering Graph Representation Learning with Test-Time Graph Transformation"
_ICLR.cc/2023/Conference — ICLR 2023 poster_

### Official Review · Reviewer_7RMb · 2022-10-24

**Confidence:** 5
**Correctness:** 3
**Technical Novelty And Significance:** 3
**Empirical Novelty And Significance:** 2
**Recommendation:** 8

**Clarity, Quality, Novelty And Reproducibility:**

Clarity: Mostly clear, but some improvements could be made.

Quality:  Generally high quality, with a lot in the appendix.

Originality:  There's recently been a lot of work in the OOD/transfer learning space with GNNs.  The paper does a pretty good job of positioning itself, but there is room for improvement.

Reproducibility:  There's no supplementary material, so this is hard to judge.  The authors promise to release the code however, and I don't think this method is too hard to implement.

**Strength And Weaknesses:**

Strengths:
+ Interesting problem setting
+ Nice experimental evaluation on a variety of real datasets.
+ Theory almost convinces me.

Weaknesses
- All experiments are on datasets which use manually created distribution shifts.  The scale of the shifts (and therefore the difficulty of the task) are impossible to ascertain without reporting more about these tasks.  For instance, the distributional shift could be quantized for each dataset (per task).
- One method which seemingly might work well in this scenario (SR-GNN) is suspiciously avoided.  But I think the method (or some similar domain shift regularization) would make a very natural baseline, and illustrate the difference between the case where the test is available (for SR-GNN) and when it is not (this paper).
- Another method which assumes there is test data available (AirGNN-t) is similarly not
- Theorem 1 doesn’t provide much value (and I can’t believe it's unique to this paper -- it's just a general statement about learning with a surrogate loss).  Might I suggest citing some sources for this here?
- I wonder if some more baselines could be added (even simpler ones).  This would add to the result.

Minor comments
- I would love a stronger emphasis on the connection between Theorem 1 and 2.  I’m not sure what to do here, but I had to revisit this a couple times.

- $\Delta A$ just feels hard to read, could you use a subscript or something? (e.g. $\Delta_A$ or $A_\Delta$)

- There's some missing related work on graph structure learning (all of which makes the same sparsity assumption as this work iirc)

  - “Grale: Designing networks for graph learning” - Halcrow et al
  - “Pathfinder Discovery Networks for Neural Message Passing” - Rozemberczki et al
  - "SLAPS: Self-Supervision Improves Structure Learning for Graph Neural Networks" - Fatemi et al


**Summary Of The Paper:**

This paper proposes a method which can adapt a testing graph for best performance with some pretrained GNN model.  It finds this change to the test graph by jointly optimizing the feature and adjacency matrices via projected gradient descent to maximize a self-supervised loss (basically drop edge + DGI) over the test graph itself.  Extensive experimental results are performed some synthetic distribution shift scenarios to illustrate the model's capabilities.

The paper seems like it could potentially be a nice contribution, but it could use a little refinement.


**Summary Of The Review:**

The authors have addressed the majority of my concerns and provided substantially more evidence to support their method.

---

> ### Author Response · Authors · 2022-11-18
> **Response to Reviewer 7RMb - Part 1**
>
> We appreciate the reviewer's perception of our contributions and thank the reviewer for the insightful questions. Our detailed responses are below.
>
> > Q1. All experiments are on datasets which use manually created distribution shifts. The scale of the shifts (and therefore the difficulty of the task) are impossible to ascertain without reporting more about these tasks. For instance, the distributional shift could be quantized for each dataset (per task).
>
> A1. Following SR-GNN[1], we adopt central moment discrepancy (CMD)[2] as the measurement to quantify the distribution shifts in different graphs. We newly added the results in Table 6 in Appendix C.1 and we can observe certain distribution shifts as these values are not small. Let’s take the Ogb-arxiv dataset as an example, where we select papers published before 2011 for training, 2011-2014 for validation, and within 2014-2016/2016-2018/2018-2020 for test. In this context, the distribution shift is from the temporal change.  In the following table, we show the CMD values, ERM performance and GTrans performance.  From the table, we can find that (1) the CMD value on the validation graph is essentially smaller than those on test graphs; and (2) GCN performances on test graphs (with larger shifts) are lower than that on the validation graph.
>
> | Method | 2011-2014 (Val)  | ﻿2014-2016   | ﻿2014-2016   | 2016-2018   | 2018-2020   |
> |--|----|---|------|----|-------|
> | CMD    | 2.5              | 14.7        | 14.7        | 20.6        | 10.4        |
> | ERM    | 45.32+-0.50      | 41.29+-1.13 | 41.29+-1.13 | 38.69+-1.33 | 35.78+-1.81 |
> | GTrans | 45.82+-0.38      | 44.03+-0.95 | 44.03+-0.95 | 41.90+-1.28 | 38.81+-1.47 |
>
> Moreover, we show the shifts on the abnormal feature setting in the following table. We can find that a larger noise ratio would obtain a larger CMD value, indicating that the task is getting more difficult.
> | Noise ratio | 0| 0.1    | 0.2    | 0.3    | 0.4     |
> |----|---|---|---|--|---|
> | Citeseer| 0.02  | 4.69   | 8.44   | 12.87  | 15.24   |
> | Cora| 0.03 | 2.98   | 6.01   | 8.16   | 10.44   |
> | Ogb-arxiv| 11.52 | 294.91 | 578.62 | 858.85 | 1135.09 |
> | Pubmed| 0.02  | 3.24   | 6.58   | 9.14   | 10.71   |
>
> We also show the shifts on the adversarial attack setting. With the increase of perturbation rate, the CMD values also get larger.
> | Ptb. Rate | 0.00  | 0.05  | 0.10  | 0.15  | 0.20  | 0.25  |
> |---|--|---|---|--|--|--|
> | Ogb-arxiv | 11.52 | 12.56 | 12.93 | 13.13 | 13.25 | 13.42 |
>
> [1]  Shift-Robust GNNs: Overcoming the Limitations of Localized Graph Training Data. NeurIPS 2021
> [2] Central Moment Discrepancy (CMD) for Domain-Invariant Representation Learning. ICLR 2017
>
> > Q2. SR-GNN  (or some similar domain shift regularization) would make a very natural baseline. Please illustrate their difference.
>
> A2. Thank you for mentioning this important work (SR-GNN).  We did not include SR-GNN because it is originally developed under the transductive setting where the training graph and test graph are the same, while in our OOD setting the training graph and test graph are different.  We first compare their difference and then present their empirical performance. SR-GNN can be categorized as a graph domain adaptation (GraphDA), since it requires access to test data during training. We would like to highlight that our work is related to GraphDA but they are also **highly different**. The following table summarizes the key differences between our work and GraphDA.
> | **Setting**  | **Source** | **Target** | **Train Loss** | **Test Loss**     | **Parameter**  | **Efficiency** |
> |------|-------|-------|------|------|-----|------|
> | GraphDA  | $G_\text{tr}$  | $G_\text{te}$  | $\mathcal{L}(G_\text{tr}, \mathcal{Y}_\text{tr})+\mathcal{L}(G_\text{tr},   G_\text{te})$  | -                           | $f_\theta$     | Low            |
> | Gtrans   | - | $G_\text{te}$  | -    | $\mathcal{L}(G_\text{te})$  | $G_\text{te}$  | High        |
>
> In detail, there are the following differences:
> 1. **Data and losses**: GraphDA methods optimize the loss function based on both labeled source data (training data) and unlabeled target data (test data), while GTrans only requires target data during inference. Hence, GraphDA methods are infeasible when accessing the source data is prohibited such as online service.
> 2. **Parameter**: To our best knowledge, existing GraphDA methods are model-centric approaches while GTrans is a data-centric approach. GTrans adapts the data instead of the model, which can be more useful in some settings as we showed in the Example of Section 3.3.
> 3. **Efficiency**: GraphDA is indeed a training-time adaptation and for each given test graph, it would require training the model on the source and target data. Thus, it is much less efficient than GTrans, especially when we have multiple test graphs (e.g., 33 test graphs for Elliptic).
>
> [To be continued. In the next comment, we will empirically compare their performance.]

---

> > ### Author Response · Authors · 2022-11-18
> > **Response to Reviewer 7RMb - Part 2**
> >
> > To compare their empirical performance, we include two GraphDA methods (SR-GNN [1] and UDA-GCN [3]) and one general domain adaptation method (DANN [4]). SR-GNN regularizes the model’s performance on the source and target domains. Note that SR-GNN is originally developed under the transductive setting where the training graph and test graph are the same. To apply SR-GNN in our OOD setting, we assume the test graph is available during the training stage of SR-GNN, as typically done in domain adaptation methods. UDA-GCN is another work that tackles graph data domain adaptation, which exploits local and global information for different domains. In addition, we also include DANN, which adopts an adversarial domain classifier to promote the similarity of feature distributions between different domains. We followed the authors' suggestions in their paper to tune the hyper-parameters and the results are shown below.
> > | Method  | Amz-photo      | Cora           | Elliptic       | Fb100          | Ogb-arxiv      | Twitch-e       |
> > |---------|----------------|----------------|----------------|----------------|----------------|----------------|
> > | ERM     | 93.79±0.97     | 91.59±1.44     | 50.90±1.51     | 54.04±0.94     | 38.59±1.35     | 59.89±0.50     |
> > | UDA-GCN | 91.70±0.35     | 92.65±0.46     | 51.57±1.31     | 54.11±0.54     | 39.43±0.71     | 52.12±0.38     |
> > | DANN    | 94.08±0.21     | 92.89±0.64     | 53.00±0.97     | 51.53±1.47     | 36.60±1.26     | 60.13±0.53     |
> > | SR-GNN   | **94.64±0.17** | 94.08±0.28     | 51.94±0.81     | 54.08±1.10     | 38.92±0.65     | 59.21±0.51     |
> > | GTrans  | 94.13±0.77     | **94.66±0.63** | **55.88±3.10** | **54.32±0.60** | **41.59±1.20** | **60.42±0.86** |
> >
> > On the one hand, we can observe that GraphDA methods generally improve the performance of GCN under distribution shift and SR-GNN is the best baseline. On the other hand, GTrans performs the best on all datasets except Amazon-Photo. On Amazon-Photo, GTrans does not improve as much as SR-GNN, which indicates that  joint optimization over source and target is necessary for this dataset. However, recall that domain adaptation methods are less efficient due to the joint optimization on source and target: the adaptation time of SR-GNN on the 8 graphs of  Amazon-Photo is **83.5s** while that of GTrans is **4.9s** (plus pre-training time 10.1s). Overall, test-time graph transformation exhibits strong advantages of effectiveness and efficiency.
> >
> > Thank you again for the suggestions. We had included the above discussion in Appendix D.1. Overall, we believe that our work **makes unique contributions** to the community of graph machine learning, and test-time graph transformation is **a new direction that opens up new opportunities** for graph structure learning, domain generalization and adversarial robustness.
> >
> >
> > > Q3. Another method which assumes there is test data available (AirGNN-t) is similarly not
> >
> > A3. `It seems that this question is incomplete`. Would you mind taking a quick look at this issue?
> >
> > > Q4. Theorem 1 doesn’t provide much value (and I can’t believe it's unique to this paper -- it's just a general statement about learning with a surrogate loss). Might I suggest citing some sources for this here?
> >
> > A4. Thank you for the comments. We would like to clarify two points:
> > * Theorem 1 is important in that it provides theoretical insights on how to choose a suitable loss for test-time graph transformation (TTGT). Our analysis shows that a good surrogate loss should share relevant information with the downstream task (their gradients should have positive correlation). In Figure 2 ((newly added experiment), we also empirically verified this condition and showed that Eq. 6 usually shares a positive correlation with the test loss during the training trajectory.
> > * It is true that Theorem 1 is a general statement and it is indeed a **unique** contribution of this paper. In the first test-time training paper [5], it shows that positive gradient correlation between the loss functions leads to better performance on the downstream task. However, the analysis is made under strong assumptions that the model is convex and smooth, which is generally not true for non-linear graph neural networks. Another work [6] does not have such an assumption, but the analysis is more like an approximation and does not guarantee the reduction of test loss. By contrast, our Theorem 1 provides a rigorous analysis on when we can reduce the test loss without assuming the convexity or smoothness of the model.
> >
> > > Q5. I wonder if some more baselines could be added (even simpler ones). This would add to the result.
> >
> > A5. Thanks for the suggestion. Please refer to my answer in A2, where we included three more baselines about graph domain adaptation.

---

> > > ### Author Response · Authors · 2022-11-18
> > > **Response to Reviewer 7RMb - Part 3**
> > >
> > > Minor comments
> > >
> > > > Q6. I would love a stronger emphasis on the connection between Theorem 1 and 2. I’m not sure what to do here, but I had to revisit this a couple times.
> > >
> > > A6.  Thank you for the suggestion. Theorem 1 is used to guide us to find a suitable surrogate loss (contrastive loss) while Theorem 2 provides an idea why the contrastive loss helps the classification task. First, we added a new experiment to show that the gradients of the contrastive loss (Eq. 6) and test loss have a positive correlation, and optimizing it can indeed reduce the test loss. Moreover, Theorem 2 shows that minimizing Eq. 6 approximately minimizes the conditional $H(Z|Y)$, which can be rewritten as $H(Z|Y)=H(Z)-I(Z,Y)$. It indicates that minimizing Eq. 6 can also help promote $I(Z,Y)$, the mutual information between the hidden representation and downstream class. Thus, the loss in Eq. 6 actually shares relevant information with the downstream task. To clarify this point, we have included the above discussion in Section 3.3.
> > >
> > >
> > > > Q7. ΔA just feels hard to read, could you use a subscript or something?
> > >
> > > A7. Thank you for the suggestion.  We have replaced $\Delta {\bf A}$ with $\Delta_{\bf A}$ throughout the paper. Now the notations look much nicer!
> > >
> > > > Q8. There's some missing related work on graph structure learning (all of which makes the same sparsity assumption as this work iirc)
> > >
> > > A8. Thank you for the suggestions. We have cited these relevant papers in Section 2 and carefully discussed the difference between GSL and GTrans  in Appendix D.2.
> > >
> > > We would like to further compare GSL with GTrans. GSL learns the graph structure during the training time while not adapting the graph structure at test stage. Our proposed test-time graph transform is essentially different from these works as we do not modify the training data but only the test data. It can be of interest to adopt GSL method at test time by also adapting the test graph structure. However, most existing GSL methods optimize the cross entropy loss defined on the labels to update graph structure, thus not applicable in the absence of test labels. One exception is SLAPS  which utilizes a self-supervised loss together with the cross entropy loss to optimize the graph structure. However, the default setting in SLAPS is generating structure for raw data points (with no given graph structure). Hence, using SLAPS for our settings requires considerable changes. Furthermore, we highlight two more weaknesses of SLAPS compared to GTrans.
> > > * **Introducing additional parameters**. SLAPS  uses a denoising loss as self-supervision. In detail, it first injects noise into node features and trains a denoising autoencoder to denoise the noisy features. This introduces additional parameters from the denoising autoencoder and inevitably changes the model architecture.
> > > * **Not learning features**. As other GSL methods, SLAPS does not learn node features. We argue that feature learning is highly important under the abnormal feature setting as shown in Table 16. For example, structure learning only improves GCN by 2\% on \arxiv while feature learning can improve GCN by 20\%. Thus, without the feature learning component, the performance will significantly drop when encountering noisy features.
> > >
> > > Since GTrans is highly versatile and we can use any self-supervised loss as the surrogate loss, we can simply replace the contrastive loss in Eq.~(6) with the denoising loss of SLAPS instead of paying considerable efforts in adjusting SLAPS. We refer to the loss used for denoising as SLAPS loss and adopt it for TTGT. Note that we first train the parameters of the DAE used for denoising while keeping the pre-trained model fixed. Then we fix both DAE and the pre-trained model and optimize the test graph for TTGT. The results are shown in Table 11. From the table, we can observe that **the SLAPS loss (or feature denoising loss) does not work as well as the contrastive loss**.
> > >
> > > ----
> > > In light of these responses, we hope we have addressed your concerns, and hope you will consider increasing your score. If we have left any notable points of concern unaddressed, please do share and we will attend to these points.
> > >
> > > ----
> > > [1]  Shift-Robust GNNs: Overcoming the Limitations of Localized Graph Training Data. NeurIPS 2021
> > >
> > > [2] Central Moment Discrepancy (CMD) for Domain-Invariant Representation Learning. ICLR 2017
> > >
> > > [3] Unsupervised Domain Adaptive Graph Convolutional Networks. WWW 2020
> > >
> > > [4] Domain-Adversarial Training of Neural Networks. JMLR, 2016
> > >
> > > [5] Test-Time Training with Self-Supervision for Generalization under Distribution Shifts. ICML 2020
> > >
> > > [6] Improving Adversarial Defense with Self-supervised Test-time Fine-tuning. 2021

---

> > > > ### Comment · Reviewer_7RMb · 2022-12-06
> > > > **Thanks**
> > > >
> > > > Thanks for the detailed replies.  Most of my concerns have been addressed, so I will raise my score.

---

### Official Review · Reviewer_WrUj · 2022-10-25

**Confidence:** 4
**Correctness:** 3
**Technical Novelty And Significance:** 2
**Empirical Novelty And Significance:** 3
**Recommendation:** 6

**Clarity, Quality, Novelty And Reproducibility:**

The paper is mostly clear. A good amount of details have been included for reproducibility and the code has been promised to be made publicly available. Regarding novelty, refer to the first weakness I raised.

**Strength And Weaknesses:**

Strengths:
+ Good set of experiments across different tasks and with different GNNs as backbone
+ Interesting analysis of the results (I like Table 4)
+ Clarity of writing
+ Good set of related work

Weaknesses:
- The surrogate loss function used in Eq. 6 has been previously used in multiple unsupervised graph representation learning approaches and its merit has been established. Moreover, modifying the graph structure and node features has also been vastly studied in previous work, but as far as I know in all works it has been done during training. So the main novelty of the presented work seems to be in using the combination of the above two **only at the test time** (in fact at a high-level the paper could be viewed as a generalization of the SLAPS model [1] applied only at the test time). This is fine given the high performance of the model. But that makes me wonder why the approach has not been tested against other graph structure learning (GSL) approaches when GSL is done both during training, and also during testing with a fixed model (only the structure changing)? This could have been done with SLAPS [1], IDGL [2], LDS [3], etc.

- \Delta A has been modelled as a full matrix of parameters making the approach not scalable to large graphs.

- I was quite excited when I read theorem 2 but when I looked at the proof, I noticed that approximation between Eq. 17 and Eq. 19 is so coarse that I'm not sure if it even means much. While Eq. 17 requires that each node has a similar embedding across different views, Eq. 19 requires that the two embeddings for each pair of nodes within the same class have similar embeddings; the only connection between the two seems to that both embeddings of the same node should belong to the same class which is a very weak connection.

- I found the example in 3.3 to be somewhat weak: while in theory one can modify the data so the model makes correct predictions for x_1 and x_2, in practice the modification is based on some predefined procedure (in the case of the current paper, the procedure being the optimization of Eq 6) so I would expect the data modifications will still result in the model making the same prediction for both nodes.

- Some of the improvements in Table 1 are not statistically significant so the "Rank" column is misleading.

Question:
- It was not clear to me why the alternating optimizations are used. Why not optimize both losses at the same time?

- How do you decide how many gradient descent steps to use at the test time?

[1] SLAPS: Self-Supervision Improves Structure Learning for Graph Neural Networks
[2] Iterative Deep Graph Learning for Graph Neural Networks: Better and Robust Node Embeddings
[3] Learning Discrete Structures for Graph Neural Networks

**Summary Of The Paper:**

This paper proposes a data-centric approach for improving the test set performance of GNNs under distribution shift, abnormal features, and adversarial attacks. In particular, the approach assumes the existence of a pre-trained GNN model and aims at improving the test set performance by updating the graph structure and node attributes at the test time with a surrogate loss function. Some theoretical results have been provided to explain why the proposed approach works. A good set of experimental results have been also provided across different setups to show the benefit of the approach empirically.

**Summary Of The Review:**

The paper shows strong results with a combination of two existing techniques (graph structure learning, and a contrastive surrogate loss function for self-supervised learning) during test time. While the empirical aspect of the paper is strong (expect for some possible GSL baselines), the theoretical aspect (and intuitions) are somewhat lagging.

---

> ### Author Response · Authors · 2022-11-18
> **Response to Reviewer WrUj - Part 1**
>
> We appreciate the reviewer's perception of our contributions and thank the reviewer for the insightful questions. Our detailed responses are below.
> > Q1. The surrogate loss function used in Eq. 6 has been previously used in multiple unsupervised graph representation learning approaches and its merit has been established.
>
> A1. Thank you for the comment. We would like to draw your attention to the following points:
> * Firstly, our core contribution in the surrogate loss section (Section 3.2) does not lie in the parameter-free surrogate loss. Instead, we aim to **provide theoretical insights** on how to choose a suitable loss for test-time graph transformation (TTGT) in the absence of test labels. Our analysis shows that a good surrogate loss should share relevant information with the downstream task (their gradients should have positive correlation). Our (newly added) study in Figure 2 also **supports this claim empirically**. Such analysis inspires us to investigate the potential of contrastive learning in the TTGT problem, since graph contrastive learning has shown enormous success in the node classification task.
> * However, as the majority of graph contrastive learning methods require a projection head to transform the hidden representations [1,2,3,4,5,6], we argue that it is necessary to remove the projection head in test-time graph transformation so as to keep the pre-trained architecture unchanged.
> * The proposed GTrans is highly composable in that we can try any self-supervised losses (or contrastive loss) but remove the additional parameters of projection. For example, in Appendix D.9, we examined three more self-supervised losses and found that the contrastive learning loss works the best for TTGT.
>
> > Q2. Modifying the graph structure and node features has also been vastly studied in previous work (it has been done during training). So the main novelty of the presented work seems to be in using the combination of the above two only at the test time (in fact at a high-level the paper could be viewed as a generalization of the SLAPS model applied only at the test time).
>
> A2. Thank you for the comments. We would like to clarify the novelty and technical contributions of our work.
> * We are the first to propose the framework of test-time graph transformation (TTGT), which bridges the gap between test graph and training graph given the pre-trained model. While we employ techniques from GSL and unsupervised learning to complete this framework, we make non-trivial contributions on adapting them.
> * Compared to the works in graph contrastive learning, we aim to provide theoretical analysis on what surrogate loss we should choose for TTGT and why contrastive loss can help us learn good solutions for this problem.
> * In comparison to GSL, our work is relevant to this research direction but also have many key differences. we encourage the reviewer to take a look at the detailed answer in Q3.
>
> > Q3. Why the approach has not been tested against other graph structure learning (GSL) approaches when GSL is done both during training, and also during testing with a fixed model (only the structure changing)? .
>
> A3.  We did not include GSL methods in the baselines because they **do not specifically tackle the distribution shift issue**.  Our proposed test-time graph transform is **essentially different** from these works as we do not modify the training data but only the test data. It can be of interest to adopt GSL method at test time by also adapting the test graph structure. However, most existing GSL methods optimize the cross entropy loss defined on the labels to update graph structure, thus not applicable in the absence of test labels. One exception is SLAPS which utilizes a self-supervised loss together with the cross entropy loss to optimize the graph structure. However, the default setting in SLAPS is generating structure for raw data points (with no given graph structure). Hence, using SLAPS for our settings *requires considerable changes*. Furthermore, we highlight two more weaknesses of SLAPS compared to GTrans.
>
> * **Introducing additional parameters**. SLAPS  uses a denoising loss as self-supervision. In detail, it first injects noise into node features and trains a denoising autoencoder to denoise the noisy features. This introduces additional parameters from the denoising autoencoder and inevitably changes the model architecture.
> * **Not learning features**. As other GSL methods, SLAPS does not learn node features. We argue that feature learning is highly important under the abnormal feature setting as shown in Table 16. For example, structure learning only improves GCN by 2\% on \arxiv while feature learning can improve GCN by 20\%. Thus, without the feature learning component, the performance will significantly drop when encountering noisy features.
>
> [To be continued. In the next comment, we will show how we combine GTrans with SLAPs.]

---

> > ### Author Response · Authors · 2022-11-18
> > **Response to Reviewer WrUj - Part 2**
> >
> > Since GTrans is highly versatile and we can use any self-supervised loss as the surrogate loss, we can simply replace the contrastive loss in Eq. (6) with the denoising loss of SLAPS instead of paying considerable efforts in adjusting SLAPS. We refer to the loss used for denoising as SLAPS loss and adopt it for TTGT. Note that we first train the parameters of the DAE used for denoising while keeping the pre-trained model fixed. Then we fix both DAE and the pre-trained model and optimize the test graph for TTGT. The results are shown in the following table. From the table, we can observe that the **SLAPS loss (or feature denoising loss) does not work as well as the contrastive loss**. In Figure 9(c) (Page 26), we also show that the gradients of SLAPS loss do not have a positive correlation with the test loss, thus not helping the downstream task, which **demonstrates the importance of Theorem 1**.
> > | None       | Amz-photo  | Cora       | Elliptic   | Fb100      | Ogb-arxiv  | Twitch-e   |
> > |------------|------------|------------|------------|------------|------------|------------|
> > | ERM        | 93.79±0.97 | 91.59±1.44 | 50.90±1.51 | 54.04±0.94 | 38.59±1.35 | 59.89±0.50 |
> > | TTGT+SLAPS | 93.97±1.04 | 91.41±1.23 | 50.54±1.81 | 54.08±0.76 | 41.38±1.35 | 59.85±0.68 |
> > | GTrans     | 94.13±0.77 | 94.66±0.63 | 55.88±3.10 | 54.32±0.60 | 41.59±1.20 | 60.42±0.86 |
> >
> > Thank you again for the suggestion, which makes our paper stronger. We have included the above discussion in Appendix D.2.
> >
> > Overall, we believe that our work **makes non-trivial contributions** to the community of graph machine learning, and test-time graph transformation is **a new direction that opens up new opportunities** for graph structure learning, domain generalization and adversarial robustness.
> >
> > > Q4. \Delta A has been modeled as a full matrix of parameters making the approach not scalable to large graphs.
> >
> > A4. As we mentioned in the end of Section 3.1, we constrain the search space to only the existing edges of the graph, which is typically sparse. This strategy enables GTrans to scale up to large graphs like Ogb-arxiv (over 100k nodes) and the additional time overhead is only **2.6 seconds** and the total GPU memory is **3.9 GB** as shown in Table 2.
> >
> > > Q5.  I noticed that the approximation between Eq. 17 and Eq. 19 is coarse. While Eq. 17 requires that each node has a similar embedding across different views, Eq. 19 requires that the two embeddings for each pair of nodes within the same class have similar embeddings; the only connection between the two seems to that both embeddings of the same node should belong to the same class which is a very weak connection.
> >
> > A5. Thank you for the comments. There are two assumptions in Theorem 2: (1) data augmentation generates a data view of the same class; and (2) the means of $Z$ and $\hat{Z}$ remain the same for each class.
> > * We start from examining the first assumption. Data augmentation is widely used in defining self-supervision signals to produce multiple views of the same sample [7,8,9]. In noted in [10], “data augmentation overcomes the lack of training data by artificially inflating the training set with **label preserving transformation**”.  In recent theoretical works on understanding contrastive learning, data augmentation is also assumed to produce similar samples within the same class. For example, as noted in [9], “augmentations of data from different classes in the downstream tasks are assumed to be nearly disconnected, whereas there are more connections within the same class”. Thus, we argue that the first assumption is reasonable.
> > * On a separate note, the second assumption can be inferred by the first assumption. Let $p_k(X)$ denote the distribution of samples with class $k$ and let $x\sim p_k(X)$ denote the sample with class $k$. Recall that we assume the data augmentation function $\mathcal{A}(\cdot)$ is strong enough to generate a data view that can simulate the test data from the same class. In this regard, the new data view can be regarded as an independent sample from the same class, i.e., $\mathcal{A}(x)\sim p_k(X)$. Hence, the expectation of $Z$ and  $\hat{Z}$ is the same and we would approximately have that “the mean of ${Z}$ and that of $\hat{Z}$ are the same for each class”. Particularly, when the number of samples is relatively large, the mean of $Z$ ($\hat{Z}$) would be close to the true distribution mean. For example, on one graph of Cora, the mean absolute difference between the mean representations  of $Z$ and $\hat{Z}$ are [0.018, 0.009, 0.021, 0.024, 0.016, 0.014, 0.0, 0.016, 0.023] for each class, **which are actually very small**.
> >
> > Thank you again for the suggestions and we have included the above discussion around the proof of Theorem 2 (Page 16).

---

> > > ### Author Response · Authors · 2022-11-18
> > > **Response to Reviewer WrUj - Part 3**
> > >
> > > > Q6. I found the example in 3.3 to be somewhat weak: while in theory one can modify the data so the model makes correct predictions for x_1 and x_2, in practice the modification is based on some predefined procedure (in the case of the current paper, the procedure being the optimization of Eq (6) so I would expect the data modifications will still result in the model making the same prediction for both nodes.
> > >
> > > A6. Thank you for the comment. We agree that in practice the optimization highly depends on the choice of the surrogate loss (in this paper it is Eq. 6). As shown in Theorem 1, if the gradients of the surrogate loss have a positive correlation with the real test loss, reducing the surrogate loss can also reduce the test loss. In Figure 2 (newly added experiment), we also empirically verified this condition and showed that Eq. 6 usually shares a positive correlation with the test loss along the training trajectory. In this context, modifying the data would still reduce the test loss, with optimizing the surrogate loss, and to some point it may achieve the lowest classification error. By contrast, modifying the model can never achieve such a point.
> > >
> > > > Q7. Some of the improvements in Table 1 are not statistically significant so the "Rank" column is misleading.
> > >
> > > A6. Thank you for the suggestions. We added significance results from paired t-tests (comparing ERM and GTrans) to justify the improvement in Table 1. From the table, we observe that most of the improvements (16/24) are significant, especially for Ogb-arxiv and Elliptic datasets. For example, when using GCN as the backbone model, GTrans improves it by a margin of **5.0% and 3.0%** on Ogb-arxiv and Elliptic, respectively.  Moreover, we note that GTrans generally **does not degrade the performance**, while other baselines such as EERM would actually show certain performance degradation in a few datasets.  As for the Rank column, it shows the average rank of these methods on all datasets, and we believe that it can generally give us a hint of how well these methods behave.
> > >
> > > > Q7. It was not clear to me why the alternating optimizations are used. Why not optimize both losses at the same time?
> > >
> > > A7. Alternating optimization is a popular technique in machine learning [11,12,13], which updates two sets of variables in an alternating manner. Particularly, [13] shows that alternating optimization of features and structure can work better than jointly optimizing them. Another reason we adopt this alternating strategy is that it gives more flexibility for balancing the learning of features and structure (as we can tune the hyper-parameters of their optimization epochs).
> > >
> > >
> > > > Q8. How do you decide how many gradient descent steps to use at the test time?
> > >
> > > A8. Following the suggestions by the authors of test-time training [14], we search the hyper-parameters based on the validation performance. Specifically, we adopt GTrans to transform validation graphs and select the hyper-parameters that lead to the best validation performance. In our experiments, the number of steps is searched in the range of {5, 10}.
> > >
> > >
> > > ----
> > > In light of these responses, we hope we have addressed your concerns, and hope you will consider increasing your score. If we have left any notable points of concern unaddressed, please do share and we will attend to these points.
> > >
> > > ----
> > > [1] Deep Graph Infomax. ICLR 2019
> > >
> > > [2] InfoGraph: Unsupervised and Semi-supervised Graph-Level Representation Learning via Mutual Information Maximization. ICLR 2020
> > >
> > > [3] Graph Contrastive Learning with Augmentations. NeurIPS 2020.
> > >
> > > [4] Graph Contrastive Learning with Adaptive Augmentation. WWW 2021.
> > >
> > > [5] ADGCL : Adversarial Graph Augmentation to Improve Graph Contrastive Learning. NeurIPS 2021.
> > >
> > > [6] An Empirical Study of Graph Contrastive Learning. NeurIPS 2021.
> > >
> > > [7] Momentum contrast for unsupervised visual representation learning. ICCV 2020.
> > >
> > > [8] A simple framework for contrastive learning of visual representations.ICML  2020
> > >
> > > [9] Provable Guarantees for Self-Supervised Deep Learning with Spectral Contrastive Loss. NeurIPS 2021
> > >
> > > [10] Improving Deep Learning using Generic Data Augmentation. 2017
> > >
> > > [11] Joint Optimization Framework for Learning with Noisy Labels. CVPR 2018
> > >
> > > [12] Graph Structure Learning for Robust Graph Neural Networks. KDD 2020
> > >
> > > [13] Graph Condensation for Graph Neural Networks. ICLR 2022
> > >
> > > [14] Test-Time Training with Self-Supervision for Generalization under Distribution Shifts. ICML 2020

---

> > > > ### Comment · Reviewer_WrUj · 2022-12-06
> > > > **Reviewer Response**
> > > >
> > > > Thanks for the detailed responses. The response addresses my concerns regarding the novelty and empirical performance. But as mentioned in the original review, my main concern was regarding the theoretical analyses (and the intuitions provided using the examples) which are still not addresses.
> > > >
> > > > For Theorem 2, the response is only focusing on justifying the assumptions, whereas my concern is regarding the approximation. To be more specific, the goal is to optimize Eq (19) and Eq (17) is argued to be an approximation of it. Let's focus only on a single class and assume $\mathbf{S}_{ij}=-\hat{\mathbf{z}}_i^T \mathbf{z}_j$. The objective of Eq (19) is to minimize the sum of all of the elements of $\mathbf{S}$ whereas Eq (17) only sums the diagonal of the matrix. Is this correct?
> > > > If yes, how can the sum of the diagonal elements of a matrix be an approximation to the sum of all the elements of the matrix? In the words of the authors: "[...] the only difference is that Eq. (19) includes more positive pairs for loss calculation" but this is only valid if the subsampled elements are selected randomly, not when the diagonal is selected.
> > > > (Unless there is a mistake in my understanding explained above, in my humble opinion the Theorem is better removed the paper).

---

> > > > > ### Author Response · Authors · 2022-12-06
> > > > > **Response to the followup question**
> > > > >
> > > > > Thank you for the feedback and we are glad that we addressed your other concerns. In the following, we will clarify your concern on the approximation of Eq (19) with Eq (17).
> > > > >
> > > > > >  In the words of the authors: "[...] the only difference is that Eq. (19) includes more positive pairs for loss calculation" but this is only valid if the subsampled elements are selected randomly, not when the diagonal is selected.
> > > > >
> > > > > We agree that this is only valid if the subsampled elements are selected randomly; and we argue that our subsampled elements are indeed randomly sampled instead of only sampled from the diagonal.
> > > > >
> > > > > Let's recall the following statement in our previous response (the validation for the assumption in A5).
> > > > >
> > > > > >  Let $p_k(X)$ denote the distribution of samples with class $k$ and let $x\sim p_k(X)$ denote the sample with class $k$. Recall that we assume the data augmentation function $\mathcal{A}(\cdot)$ is strong enough to generate a data view that can simulate the test data from the same class. In this regard, the new data view can be regarded as an independent sample from the same class, i.e., $\mathcal{A}(x)\sim p_k(X)$.
> > > > >
> > > > > From the above statement, we show that the new data view can be regarded as an independent sample from the same class. Thus, in Eq (19), $Z_k$ can be considered the same as $\hat{Z}_k$ and ${\bf \hat{z}}_j \in {Z}_k$. Then we rewrite Eq (19) as
> > > > >
> > > > > ${\sum_{{\bf z}_i\in{} {Z}_k, \hat{\bf z}_j \in Z_k}} ( - \hat{\bf z}^\top_j {\bf z}_i )$, while Eq (17) is
> > > > >
> > > > > $\sum_{{\bf z}_i\in Z_k} \left(-\hat{\bf z}^{\top}_i {\bf z}_i\right)$. Then we need to examine the process of generating $\hat{\bf z}$: we applied **stochastic graph data augmentation**, such as randomly dropping edges from the graph, to generate new samples. According to our assumption that the new data view can be regarded as an independent sample from the same class, this augmentation process can be viewed as **randomly** drawing samples from the corresponding class and ${\bf \hat{z}}_j$ is randomly drawn from ${Z}_k$. Thus, we argue that it is reasonable to approximate Eq (19) with Eq (17).
> > > > >
> > > > > We apologize for the confusion and we will clarify it during later revision.
> > > > >
> > > > > Best,
> > > > > Authors

---

### Official Review · Reviewer_2PUQ · 2022-10-27

**Confidence:** 4
**Correctness:** 3
**Technical Novelty And Significance:** 2
**Empirical Novelty And Significance:** 3
**Recommendation:** 3

**Clarity, Quality, Novelty And Reproducibility:**

The paper is well written. The proposed approach is somewhat novel. Several technical contributions should be further elaborated. No data or code is provided to examine the reproducibility.

**Strength And Weaknesses:**

Strengths:
+ This paper proposes a novel idea that optimizes the graph structure and features of the test graph.
+ The writing is satisfactory and the idea is clear to follow.

Weaknesses:
- Several claims are not fully substantiated.
  1. Theorem 1 gives a criterion on selecting the surrogate objective while no empirical nor theoretical evidence is given to show that the contrastive learning (Eq. 6) approximates the classification loss. It could be better to show, for example on one dataset that the gradient paths of the contrastive learning indeed are close to that of the classification objective.
  2. Theorem 2 relies on very strong assumptions that the mean of $Z$ and $\hat{Z}$ remain the same for each class. Can authors further justify that this assumption is realistic?
  3. For Eq. (6) the authors claim that existing contrastive learning objectives need extra prediction head that will alter the network architecture. It is not true. For example, the JSD loss and the triplet margin loss do not include additional parameters. Also, it is possible to use bootstrapping-based loss to update the graph structure.
  4. There are also several graph contrastive learning approaches that leverage learnable graph augmentation, which should be considered as baselines as well. For example Adversarial Graph Augmentation to Improve Graph Contrastive Learning, NeurIPS 2021
  Considering the above points, I believe the third contribution needs further elaboration. The technical contribution for the parameter-free surrogate objective is undermined.
- From Table 1, it is very hard to say that GTrans achieves significantly better performance over baselines. Almost all performance gains fall into one sigma. Also, I am curious about how to select and how sensitive the model to the parameter B as it controls the overall degree of graph transformation.


**Summary Of The Paper:**

This paper presents a test-time graph transformation approach to improve graph representation learning performance. Specifically, during test time, the structure and features of the test graph will be optimized through a contrastive learning objective. Experiments on node classification under various settings demonstrate the effectiveness of this data-centric approach over model adaptation approach.


**Summary Of The Review:**

Overall I like the idea of employing test-time transformation to improve graph representation learning performance. However, there are some technical contributions that are slightly overclaimed. Experiments also only show very marginal improvements. Therefore, I recommend rejection at this time.

---

> ### Author Response · Authors · 2022-11-18
> **Response to Reviewer 2PUQ - Part 1**
>
> Thank you for your detailed comments and constructive suggestions. To address your concerns, we provide the following responses.
>
> > Q1. Theorem 1 gives a criterion on selecting the surrogate objective while no empirical nor theoretical evidence is given to show that the contrastive learning (Eq. 6) approximates the classification loss. It could be better to show, for example on one dataset that the gradient paths of the contrastive learning indeed are close to that of the classification objective.
>
> A1. Thank you for the valuable suggestions. To empirically verify the effectiveness of Theorem 1, we show the gradient correlation w.r.t. training epochs. We adopt the surrogate loss in Eq. 6 as $\mathcal{L}_s$ and plot the values of $\rho(G)$ and $\mathcal{L}_c$ on one test graph in Cora in Figure 2. We can observe that a positive $\rho{(G)}$ generally reduces the test classification loss. More results on different surrogate losses can be found in Figure 9 in Appendix D.9 and similar patterns are exhibited. For example, in Figure 9(a) we show the trends of using entropy loss as the surrogate loss: the test loss generally reduces when $\rho(G)$ is positive and starts to increase after $\rho(G)$ becomes negative.
>
> Moreover, Theorem 2 shows that minimizing Eq. 6 approximately minimizes the conditional $H(Z|Y)$, which can be rewritten as $H(Z|Y)=H(Z)-I(Z,Y)$. It indicates that minimizing Eq. 6 can also help promote $I(Z,Y)$, the mutual information between the hidden representation and downstream class. Thus, the loss in Eq. 6 actually shares relevant information with the downstream task. We have included the above discussion in Section 3.3.
>
> > Q2. Theorem 2 relies on strong assumptions that the mean of $Z$ and $\hat{Z}$ remain the same for each class.
>
> A2. Thank you for the comment. We would like to note that the assumption “the mean of $Z$ and $\hat{Z}$ remain the same for each class” can be inferred by the first assumption about data augmentation. Let $p_k(X)$ denote the distribution of samples with class $k$ and let $x\sim p_k(X)$ denote the sample with class $k$. Recall that we assume the data augmentation function $\mathcal{A}(\cdot)$ is strong enough to generate a data view that can simulate the test data from the same class. In this regard, the new data view can be regarded as an independent sample from the same class, i.e., $\mathcal{A}(x)\sim p_k(X)$. Hence, the expectation of $Z$ and  $\hat{Z}$ is the same and  we would approximately have that “the mean of ${Z}$ and $\hat{Z}$ is the same for each class”.  Particularly, when the number of samples is relatively large, the mean of $Z$ ($\hat{Z}$) would be close to the true distribution mean. For example, on one graph of Cora, the mean absolute difference between the two mean representations  of $Z$ and $\hat{Z}$ are [0.018, 0.009, 0.021, 0.024, 0.016, 0.014, 0.0, 0.016, 0.023] for each class, which are actually very small. Thank you again for the suggestion and we have included the above discussion around the proof of Theorem 2 (Page 16).
>
> > Q3. For Eq. (6) the authors claim that existing contrastive learning objectives need extra prediction head that will alter the network architecture. It is not true. For example, the JSD loss and the triplet margin loss do not include additional parameters.
>
> A3. Thank you for the comments. We would like to clarify the following points:
> * We agree that some losses may not include additional parameters. However, the majority of graph contrastive learning methods would require a projection head to transform the hidden representations [1,2,3,4,5,6,7]. We also kindly note that DGI[1], InfoGraph[2] and MVGRL (for node classification)[3] all used a parameterized transformation in the distance function of JSD loss. In test-time graph transformation, it is necessary to remove the projection head so as to keep the pre-trained architecture unchanged. Indeed, we can replace Eq. (6) with JSD loss (without using the parameterized transformation) and we can obtain the following results:
> | Method | Amz-photo  | Cora       | Elliptic   | Fb100      | Ogb-arxiv  | Twitch-e   |
> |----|-----|--------|-----|------|------|------|
> | ERM    | 93.79±0.97 | 91.59±1.44 | 50.90±1.51 | 54.04±0.94 | 38.59±1.35 | 59.89±0.50 |
> | Eq.(6) | 94.13±0.77 | 94.66±0.63 | 55.88±3.10 | 54.32±0.60 | 41.59±1.20 | 60.42±0.86 |
> | JSD    | 94.28±0.73 | 94.89±0.60 | 56.32±3.03 | 54.29±0.59 | 40.95±1.14 | 60.39±0.77 |
>
> From the table, we can observe that using the JSD in the TTGT framework can also work well and achieve comparable performance as the original contrastive loss.
>
> [To be continued on next comment...]

---

> > ### Author Response · Authors · 2022-11-18
> > **Response to Reviewer 2PUQ - Part 2**
> >
> > * Furthermore, our core contribution in the surrogate loss section *does not lie in the parameter-free surrogate loss*. Instead, we aim to **provide theoretical insights** on how to choose a suitable loss for test-time graph transformation (TTGT). Our analysis shows that a good surrogate loss should share relevant information with the downstream task (their gradients should have positive correlation). Such analysis inspires us to investigate the potential of contrastive learning in the TTGT problem, since contrastive learning has shown enormous success in graph representation learning.
> > * The proposed GTrans is **highly composable** in that we can try any self-supervised losses but remove the additional parameters for projection. For example, in Appendix D.9, we examined three more self-supervised losses and found that they are less useful than the contrastive loss for the TTGT problem.
> >
> > We have added the above discussion in Appendix E.1.
> >
> > > Q4. Some graph contrastive learning approaches that leverage learnable graph augmentation should be considered as baselines such as AD-GCL.
> >
> > A4.  Thank you for bringing up this relevant work (AD-GCL[5]). Next, we compare our method with a graph contrastive learning method with learnable augmentation AD-GCL.  Since AD-GCL is originally designed for graph classification as a pre-training strategy, a direct empirical comparison between AD-GCL and GTrans is not easy. However, due to the flexibility of GTrans, **we can integrate it into our TTGT framework**, denoted as TTGT+Ad-GCL. Specifically, we keep the pre-trained model fixed and update the model as well as the test data using the min-max loss proposed by AD-GCL. We present the empirical results in the following table.
> > | Method        | Amz-photo      | Cora           | Elliptic       | Fb100          | Ogb-arxiv      | Twitch-e       |
> > |---------------|----------------|----------------|----------------|----------------|----------------|----------------|
> > | TTGT + AD-GCL | **94.96±0.52** | 92.38±1.35     | 54.38±2.77     | 53.81±0.87     | 39.16±0.98     | 59.78±0.65     |
> > | GTrans        | 94.13±0.77     | **94.66±0.63** | **55.88±3.10** | **54.32±0.60** | **41.59±1.20** | **60.42±0.86** |
> >
> > We can observe that TTGT+AD-GCL generally performs worse than GTrans except on \photo, which indicates that GTrans is a stronger realization of TTGT. Furthermore, we highlight some key differences between it and GTrans.
> > * **Whether changing the pre-trained model.** AD-GCL requires optimization of a min-max problem which involves parameters of graph structure and model. Thus, adopting it for TTGT would change the pre-trained model.
> > * **Whether learning node features**. AD-GCL only augments the graph structure while not learning the features. We argue that feature learning is highly important under the abnormal feature setting as shown in Table 16 in Appendix D.8. For example, structure learning only improves GCN by 2\% on Ogb-arxiv while feature learning can improve GCN by 20\%. Thus, without the feature learning component, the performance will significantly drop when encountering noisy features.
> > * **Efficiency**. According to Eq. (9) in the AD-GCL paper, it calculates the similarities between all samples within each mini-batch. When we increase the batch size, we would easily get the out-of-memory issue while a small mini-batch will slow down the learning process. As a consequence, TTGT+AD-GCL is less efficient than GTrans: the adaptation time of TTGT+AD-GCL on Ogb-arxiv is **12.7s** while that of GTrans is **2.6s**.
> >
> > Thank you again for the suggestion and we have included the above discussion in Appendix D.3 (Page 22).
> >
> > Overall, our major contribution falls in *the proposal of the test-time graph transformation framework*. This framework is **highly general** and we believe that it **opens up new opportunities** for graph structure learning, self-supervised learning and domain generalization.
> >
> > > Q5. From Table 1, it is very hard to say that GTrans achieves significantly better performance over baselines.
> >
> > A5.  Thank you for the comment. We provide significance results from paired t-tests (comparing ERM and GTrans) to justify the improvement in Table 1. From the results, we observe that 16/24 of the improvements are significant, especially for Ogb-arxiv and Elliptic datasets. For example, when using GCN as the backbone model, GTrans improves it by a margin of **5.0%** and **3.0%** on Ogb-arxiv and Elliptic, respectively.  Moreover, we note that (1) GTrans generally *does not degrade the performance*, while other baselines such as EERM would actually show certain performance degradation in a few datasets; (2) The average rank of  GTrans is always **the best** in all cases, which demonstrates its superiority over other methods; and (3) GTrans is **more memory- and time-efficient** as shown in Table 2 (Page 7).

---

> > > ### Author Response · Authors · 2022-11-18
> > > **Response to Reviewer 2PUQ - Part 3**
> > >
> > > > Q6. Also, I am curious about how to select and how sensitive the model to the parameter B as it controls the overall degree of graph transformation.
> > >
> > > * When searching for hyper-parameters, we adopt GTrans to transform validation graphs and select the hyper-parameters that lead to the best validation performance.  This strategy is suggested by the authors of TTT [9].
> > > * We examine the sensitivity of GTrans' performance with respect to the perturbation budget, i.e., hyper-parameter $B$. Specifically, we vary the value of $B$ in the range of {0.5%, 1\%, 5\%, 10\%, 20\%, 30\%\} and perform experiments on the \arxiv dataset for the three settings in the following table. Specifically, "Abn. Feat" stands for abnormal feature setting with 30\% noisy feature while "Adv. Attack" stands for the adversarial attack setting with 20\% perturbation rate.
> > > | Setting    | 0.5%  | 1%    | 5%    | 10%   | 20%   | 30%   |
> > > |------------|-------|-------|-------|-------|-------|-------|
> > > | OOD        | 40.52 | 40.69 | 41.32 | 41.40 | 41.70 | 41.65 |
> > > | Abn. Feat. | 64.78 | 64.80 | 64.64 | 64.57 | 64.57 | 64.57 |
> > > | Attack     | 61.39 | 61.31 | 62.18 | 63.19 | 64.62 | 65.16 |
> > >
> > > From the table, we can observe that budget $B$ has a smaller effect on OOD and abnormal feature settings while highly impacting the performance under structural adversarial attack. This is because most of the changes made by adversarial attack are edge injections and we need to use a large budget $B$ to remove  adversarial patterns. By contrast, GTrans is much less sensitive to the value of $B$ in the other two settings.
> > >
> > >
> > > ----
> > > In light of these responses, we hope we have addressed your concerns, and hope you will consider increasing your score. If we have left any notable points of concern unaddressed, please do share and we will attend to these points.
> > >
> > > ---
> > >
> > > [1] Deep Graph Infomax. ICLR 2019
> > >
> > > [2] InfoGraph: Unsupervised and Semi-supervised Graph-Level Representation Learning via Mutual Information Maximization. ICLR 2020
> > >
> > > [3] Graph Contrastive Learning with Augmentations. NeurIPS 2020.
> > >
> > > [4] Graph Contrastive Learning with Adaptive Augmentation. WWW 2021.
> > >
> > > [5] ADGCL : Adversarial Graph Augmentation to Improve Graph Contrastive Learning. NeurIPS 2021.
> > >
> > > [6] An Empirical Study of Graph Contrastive Learning. NeurIPS 2021.
> > >
> > > [7] Contrastive Multi-View Representation Learning on Graphs. ICML 2020
> > >
> > > [8] An Empirical Study of Graph Contrastive Learning. NeurIPS 2021.
> > >
> > > [9] Test-Time Training with Self-Supervision for Generalization under Distribution Shifts. ICML 2020

---

### Official Review · Reviewer_3Tit · 2022-10-28

**Confidence:** 3
**Correctness:** 4
**Technical Novelty And Significance:** 3
**Empirical Novelty And Significance:** 3
**Recommendation:** 8

**Clarity, Quality, Novelty And Reproducibility:**

Clarity.
The  paper is well written and easy to follow.

Quality.
The paper proposes a new method to empower the graph representation learning with test-time graph transformation.  Aslo, this paper provides theoretical analysis on the design of the framework. Furthermore, the paper conducts extensive experiments on several datasets and shows better performance compared with several baselines.

Novelty.
The idea is interesting and novel. Most of prior work focuses on to improve graph representations via model structure design. Working on the test-time graph is new direction. This paper proposes a new method in this new direction.




**Strength And Weaknesses:**

Strength.
1. The research is novel and interesting.
2. The paper is well-written and easy to follow.

Weakness.
1. This paper does not provide runnable code to reproduce the result.

**Summary Of The Paper:**

GNN is a new technique that can be used in many areas such as recommendations, drug discovery, social computing, etc. How to learn the  graph representation is key issue in the area of GNNs. This paper proposes a new method to empower the graph representation learning with test-time graph transformation.  Aslo, this paper provides theoretical analysis on the design of the framework and discusses why adapting graph data works better than adapting the model. Extensive experiments on several datasets show its effectiveness.

**Summary Of The Review:**

GNN is a new technique that can be used in many areas such as recommendations, drug discovery, social computing, etc. How to learn the  graph representation is key issue in the area of GNNs. This paper proposes a new method to empower the graph representation learning with test-time graph transformation.  Aslo, this paper provides theoretical analysis on the design of the framework and discusses why adapting graph data works better than adapting the model. Extensive experiments on several datasets show its effectiveness.

Most of prior work focuses on to improve graph representations via model structure design. Working on the test-time graph is new direction. This paper proposes a new method in this new direction, which will be useful to resolve the problem of dynamic graph learning.

---

> ### Author Response · Authors · 2022-11-18
> **Response to Reviewer 3Tit**
>
> We gratefully appreciate your time in reviewing our paper, your insightful comments and your support! Your positive feedback is very encouraging for us. Below we would like to clarify your major concern.
>
> > Q. This paper does not provide runnable code to reproduce the result.
>
> A. Thank you for the suggestion. We have provided our code in the supplementary file to ensure reproducibility. A `README.md` file is included in the repository to guide the users to install necessary packages and download datasets. Following the instruction in this file, the users are able to run the experiments under the settings of out-of-distribution, abnormal features and adversarial attacks. In addition, the details for dataset statistics and hyper-parameters setup for three settings are included in Appendix C of the paper.

---

### Official Review · Reviewer_aTyX · 2022-11-05

**Confidence:** 4
**Correctness:** 3
**Technical Novelty And Significance:** 2
**Empirical Novelty And Significance:** 2
**Recommendation:** 6

**Clarity, Quality, Novelty And Reproducibility:**

The paper is overall well written and easy to follow. Currently, I think the setting is quite similar to GraphDA which limits the technical contribution. However, if the authors can well distinguish this paper with GraphDA, I will reconsider the novelty. The reproducibility is good as the authors provide detailed setting in the appendix.


**Strength And Weaknesses:**

**Strength**
- The motivation of this paper is interesting, the authors consider several critical problems in the test time of GNN, including OOD, feature anomaly and adversarial attack.
- The paper is overall easy to follow.
- The experiments are extensive.

**Weakness**
- The overall problem setting is quite similar to the graph domain adaptation(GraphDA) where the source domain graph data provides a pre-trained model and target domain is the test graph without labels. However, the authors do not mention this active research field as well as the baseline methods. In the GraphDA field, the transformation on the target domain is a quite common operation. Concerning the Equation (6), it is directly computed on the pre-trained model and serves as a weight/guidance on the trasnformation learning, this is an explicit domain adaptation step where the knowledge from the source domain is transformed to the target domain. The authors are required to clarify the difference between this paper and GraphDA task.
- The so-called parameter free transformation is quite common in contrastive learning. In this paper, the dropedge is only a classic one and many other transformations such as feature shuffling, subgraph sampling can be explored.
- The Theorem 1. is somehow weired, in the test time the graph is unlabelled and the classification loss is no longer appliable. In this situation, will theorem 1 affect the model training and transformation selection?
- In the OOD setting, the improvements is not significant and the authors are expected to explain on it, especailly without comparing the GraphDA methods.
- In the nosiy feature setting, the proposed method only works on the small dataset and seems not working on the larger dataset, the authors are expected to discuss on it.


**Summary Of The Paper:**

This paper aims at improving the performance of GNN in the test time by transferring the node features and structure.
More specifically, the transformation is optimized by a contrastive surrogate loss with respect to the embedding from the pretrained model.
Extensive experiments are conducted on different settings and datasets, the results somehow prove the effectiveness of the proposed method.

**Summary Of The Review:**

My major concerns is the difference between the problem setting and the GraphDA methods.
Currently, I think they are quite similar so the technical contribution and baselines are somehow weak to me.
However, if the authors can well address their difference, I will improve the final rating by reconsidering the technical contribution.

---

> ### Author Response · Authors · 2022-11-18
> **Response to Reviewer aTyX  - Part 1**
>
> Thank you for your detailed comments and constructive suggestions. To address your concerns, we provide the following responses.
> > Q1. The authors are required to clarify the difference between this paper and GraphDA task.
>
> A1. Thank you for bringing up the important research of graph domain adaptation (GraphDA). We would like to highlight that our work is related to GraphDA but they are also **highly different**. The following table summarizes the key differences between our work and GraphDA.
> | **Setting**  | **Source**     | **Target**     | **Train Loss** | **Test Loss**     | **Parameter**  | **Efficiency** |
> |------|-------|-------|------|------|-----|------|
> | GraphDA  | $G_\text{tr}$  | $G_\text{te}$  | $\mathcal{L}(G_\text{tr}, \mathcal{Y}_\text{tr})+\mathcal{L}(G_\text{tr},   G_\text{te})$  | -                           | $f_\theta$     | Low            |
> | Gtrans   | -              | $G_\text{te}$  | -    | $\mathcal{L}(G_\text{te})$  | $G_\text{te}$  | High        |
>
> In detail, there are the following differences:
> 1. **Data and losses**: GraphDA methods optimize the loss function based on both labeled source data (training data) and unlabeled target data (test data), while GTrans only requires target data during adaptation. Hence, GraphDA methods are infeasible when accessing the source data is prohibited such as online service.
> 2. **Parameter**: To our best knowledge, existing GraphDA methods are model-centric approaches while GTrans is a data-centric approach. GTrans adapts the data instead of the model, which can be more useful in some settings as we showed in the Example of Section 3.3.
> 3. **Efficiency**: GraphDA is indeed a training-time adaptation and for each given test graph, it would require training the model on the source and target data. Thus, it is much less efficient than GTrans, especially when we have multiple test graphs (e.g., 33 test graphs for Elliptic).
>
> To compare their empirical performance, we include two GraphDA methods (SR-GNN [1] and UDA-GCN [2]) and one general domain adaptation method (DANN [3]). SR-GNN regularizes the model’s performance on the source and target domains. Note that SR-GNN is originally developed under the transductive setting where the training graph and test graph are the same. To apply SR-GNN in our OOD setting, we assume the test graph is available during the training stage of SR-GNN, as typically done in domain adaptation methods. UDA-GCN is another work that tackles graph data domain adaptation, which exploits local and global information for different domains. In addition, we also include DANN, which adopts an adversarial domain classifier to promote the similarity of feature distributions between different domains. We followed the authors' suggestions in their paper to tune the hyper-parameters and the results are shown below.
> | Method  | Amz-photo      | Cora           | Elliptic       | Fb100          | Ogb-arxiv      | Twitch-e       |
> |---------|----------------|----------------|----------------|----------------|----------------|----------------|
> | ERM     | 93.79±0.97     | 91.59±1.44     | 50.90±1.51     | 54.04±0.94     | 38.59±1.35     | 59.89±0.50     |
> | UDA-GCN | 91.70±0.35     | 92.65±0.46     | 51.57±1.31     | 54.11±0.54     | 39.43±0.71     | 52.12±0.38     |
> | DANN    | 94.08±0.21     | 92.89±0.64     | 53.00±0.97     | 51.53±1.47     | 36.60±1.26     | 60.13±0.53     |
> | SR-GNN   | **94.64±0.17** | 94.08±0.28     | 51.94±0.81     | 54.08±1.10     | 38.92±0.65     | 59.21±0.51     |
> | GTrans  | 94.13±0.77     | **94.66±0.63** | **55.88±3.10** | **54.32±0.60** | **41.59±1.20** | **60.42±0.86** |
>
> On the one hand, we can observe that GraphDA methods generally improve the performance of GCN under distribution shift and SR-GNN is the best baseline. On the other hand, GTrans performs the best on all datasets except Amazon-Photo. On Amazon-Photo, GTrans does not improve as much as SR-GNN, which indicates that  joint optimization over source and target is necessary for this dataset. However, recall that domain adaptation methods are less efficient due to the joint optimization on source and target: the adaptation time of SR-GNN on the 8 graphs of  Amazon-Photo is **83.5s** while that of GTrans is **4.9s** (plus pre-training time 10.1s). Overall, test-time graph transformation exhibits strong advantages of effectiveness and efficiency.
>
> We have included the above discussion in Appendix D.1. Overall, we believe that our work **makes unique contributions** to the community of graph machine learning, and test-time graph transformation is **a new direction** that opens up new opportunities for graph structure learning, domain generalization and adversarial robustness.
>
> [1] Shift-Robust GNNs: Overcoming the Limitations of Localized Graph Training Data. NeurIPS 2021
>
> [2] Unsupervised Domain Adaptive Graph Convolutional Networks. WWW 2020
>
> [3] Domain-Adversarial Training of Neural Networks. JMLR, 2016

---

> > ### Author Response · Authors · 2022-11-18
> > **Response to Reviewer aTyX - Part 2**
> >
> > > Q2. The parameter-free transformation is common in contrastive learning.
> >
> > A2. Thank you for the comment. We would like to draw your attention to the following points:
> > * Firstly, our core contribution in the surrogate loss section (Section 3.2) does not lie in the parameter-free surrogate loss. Instead, we aim to **provide theoretical insights** on how to choose a suitable loss for test-time graph transformation (TTGT) in the absence of test labels. Our analysis shows that a good surrogate loss should share relevant information with the downstream task (their gradients should have positive correlation). Our (newly added) study in Figure 2 also **supports this claim empirically**. Such analysis inspires us to investigate the potential of contrastive learning in the TTGT problem, since graph contrastive learning has shown enormous success in the node classification task.
> > * However, as the majority of graph contrastive learning methods require a projection head to transform the hidden representations [4,5,6,7,8,9], we argue that it is necessary to remove the projection head in test-time graph transformation so as to keep the pre-trained architecture unchanged.
> > * The proposed GTrans is highly composable in that we can try any self-supervised losses (or contrastive loss) but remove the additional parameters of projection. For example, in Appendix D.9, we examined three more self-supervised losses and found that the contrastive learning loss works the best for TTGT.
> >
> > > Q3. DropEdge is only a classic one and many other transformations such as feature shuffling, subgraph sampling can be explored.
> >
> > A3. Thank you for pointing it out. We had chosen a popular data augmentation strategy (DropEdge) and used feature shuffling as the negative sampling strategy. Note that the augmentation used in the surrogate loss can be flexible. We would like to empirically compare different augmentations including DropEdge, node dropping and subgraph sampling, as shown in the following table. From the table, we can observe that (1) GTrans with any of the three augmentations can greatly improve the performance of GCN under distribution shift, and (2) different augmentations lead to slightly different performance on different datasets: with node dropping as the augmentation GTrans even achieves better performance than that with DropEdge in some cases.
> > | Augmentation | Amz-photo   | Cora  | Elliptic   | Fb100   | Ogb-arxiv | Twitch-e |
> > |---|---|---|----|---|---|----|
> > | Node Dropping     | **94.45±0.70** | **95.00±0.65** | **56.57±2.99** | 54.15±0.60     | 39.95±1.11     | 60.38±0.74     |
> > | Subgraph Sampling | 94.18±0.75     | 94.95±0.64     | 55.40±3.00     | **54.51±0.56** | 41.44±1.17     | **60.52±0.80** |
> > | DropEdge      | 94.13±0.77     | 94.66±0.63     | 55.88±3.10     | 54.32±0.60     | **41.59±1.20** | 60.42±0.86     |
> > | ERM               | 93.79±0.97     | 91.59±1.44     | 50.90±1.51     | 54.04±0.94     | 38.59±1.35     | 59.89±0.50     |
> >
> > > Q4. Theorem 1. is somehow weird, in the test time the graph is unlabelled and the classification loss is no longer applicable. In this situation, will theorem 1 affect the model training and transformation selection?
> >
> > A4. Thank you for the comment. We would like to clarify the following points:
> > * It is true that the graph is unlabelled at the test time and that’s why we need to design a surrogate loss (or unsupervised loss) to guide the learning process of the test graph. Given a pre-trained model, we optimize the surrogate loss defined on the unlabeled test graph to update the test graph.
> > * In the absence of test labels, Theorem 1 gives a hint of what surrogate losses are helpful in reducing the test classification loss. The conclusion is that we should choose a good surrogate loss that shares relevant information with the test classification loss. If the chosen surrogate loss satisfies this condition, optimizing the surrogate loss can lead to the reduction of test classification loss. Notably, this *does not* mean we would have the knowledge of the test labels. Given the success of graph contrastive learning, we assume that contrastive learning loss shares a certain amount of relevant information with the downstream task; and we empirically found that its gradients and the gradients of test loss do have a positive correlation in Figure 2 (newly added) in Section 3.2.
> > * Theorem 1 does not affect the model training procedure, as we keep the pre-trained model fixed and update the test graph by optimizing the surrogate loss.
> >
> > ---
> > [4] Deep Graph Infomax. ICLR 2019
> >
> > [5] InfoGraph: Unsupervised and Semi-supervised Graph-Level Representation Learning via Mutual Information Maximization. ICLR 2020
> >
> > [6] Graph Contrastive Learning with Augmentations. NeurIPS 2020.
> >
> > [7] Graph Contrastive Learning with Adaptive Augmentation. WWW 2021.
> >
> > [8] ADGCL : Adversarial Graph Augmentation to Improve Graph Contrastive Learning. NeurIPS 2021.
> >
> > [9] An Empirical Study of Graph Contrastive Learning. NeurIPS 2021.

---

> > > ### Author Response · Authors · 2022-11-18
> > > **Response to Reviewer aTyX - Part 3**
> > >
> > > > Q5. In the OOD setting, the improvement is not significant and the authors are expected to explain it, especially without comparing the GraphDA methods.
> > >
> > > A5. Thank you for the comment. We have provided the comparison between GraphDA and GTrans in A1, and we can conclude that our strategy is better from the comparison. Further, we provide significance results from paired t-tests (comparing ERM and GTrans) to justify the improvement in Table 1. From the results, we observe that 16/24 of the improvements are significant, especially for Ogb-arxiv and Elliptic datasets. For example, when using GCN as the backbone model, GTrans improves it by a margin of **5.0%** and **3.0%** on Ogb-arxiv and Elliptic, respectively.  Moreover, we note that (1) GTrans generally *does not degrade the performance*, while other baselines such as EERM would actually show certain performance degradation in a few datasets; and (2) GTrans is **more memory- and time-efficient** as shown in Table 2 (Page 7).
> > >
> > > > Q6. In the noisy feature setting, the proposed method only works on the small dataset and seems not working on the larger dataset, the authors are expected to discuss it.
> > >
> > > A6. Thanks for the comment. We argue that our method works on both small and large datasets in the noisy feature setting. We would like to clarify it from the following two points:
> > > * In Figure 3 and Figure 7, the brown plot (GTrans) is always much higher than the blue plot (GCN) on all four datasets, indicating that GTrans significantly improves the performance of the pre-trained model (GCN) by a large margin on both small and large datasets. For example, on Ogb-arxiv with 30% noisy nodes, GTrans improves GCN by **26.1% on abnormal nodes** and **20.3% on overall test accuracy**.
> > > * In Figure 3(d), GTrans performs slightly worse than AirGNNs on Ogb-arxiv with noise ratio smaller than 30% while it outperforms AirGNNs on higher noise ratios. Such observation indeed shows the difference between model-centric approaches (e.g., AirGNNs) and data-centric approaches (e.g., GTrans). As outlined in recent studies of data-centric AI, data quality is particularly important when there is a lack of sufficient high-quality data [10,11,12]. Thus, we conjecture that this is the reason why GTrans works better on small datasets (or large datasets with high noise ratio).
> > >
> > > [10] Advances, Challenges and Opportunities in Creating Data for Trustworthy AI. Nature Machine Intelligence 2022.
> > >
> > > [11] Data Excellence for AI: Why Should You Care? Interactions, 2022.
> > >
> > > [12] "Everyone wants to do the model work, not the data work": Data Cascades in High-Stakes AI. CHI 2021
> > >
> > > -----
> > > In light of these responses, we hope we have addressed your concerns, and hope you will consider increasing your score. If we have left any notable points of concern unaddressed, please do share and we will attend to these points.

---

### Author Response · Authors · 2022-11-18
**Summary of the major revision**

We thank the reviewers for the thorough and detailed reviews on our submission. We summarize major changes that we have made below. All changes are marked in blue in the updated submission.

* We provided empirical investigation on the correlation between gradients and losses (Figure 2, Page 4; Figure 9, Page 26), which supports the claim in Theorem 1.
* We clarified the key differences between graph domain adaptation (GraphDA) and our method and included their empirical comparison in Appendix D.1 (Page 21);
    - Our method is essentially different from GraphDA methods as we do not require the access of labeled data during adaptation.
    - Our method is more efficient and generally achieves better performance.
* We provided a deeper discussion on graph structure learning and demonstrated the superiority of test-time graph transformation (Appendix D.2, Page 21).
* We included the statistical significance results in Table 1 from paired t-tests and justified the improvement from our framework under OOD setting.
* We included the discussion on model sensitivity to the hyper-parameter $B$ (Appendix D.10, Page 26) and model performance when using different augmentations (Appendix D.11, Page 27).
* We provided our code in the supplementary file for reproducibility.

---

### Decision · Program_Chairs · 2023-01-20

**Decision:**

Accept: poster

**Justification For Why Not Higher Score:**

Modest novelty.

**Justification For Why Not Lower Score:**

The reviewers raised some concerns including better differentiation from graph domain adaptation(GraphDA), justification on contrastive learning to approximate the classification loss, adding graph contrastive learning approaches as baselines. The authors did a nice job addressing reviewers’ comments and adding suggested experiments. Overall, the reviewers have positive assessment after author responses.

**Metareview: Summary, Strengths And Weaknesses:**

This paper proposes a new method to empower the graph representation learning with test-time graph transformation for improving the test set performance of GNNs under distribution shift, abnormal features, and adversarial attacks. During test time, the structure and features of the test graph will be optimized through a contrastive learning objective. In particular, the approach assumes the existence of a pre-trained GNN model and aims at improving the test set performance by updating the graph structure and node attributes at the test time with a surrogate loss function. It finds this change to the test graph by jointly optimizing the feature and adjacency matrices via projected gradient descent to maximize the self-supervised loss (basically drop edge + DGI) over the test graph itself. The paper provides theoretical analysis on the design of the framework and discusses why adapting graph data works better than adapting the model. Experiments on node classification under various settings demonstrate the effectiveness of this data-centric approach over model adaptation approach. The reviewers raised some concerns including better differentiation from graph domain adaptation(GraphDA), justification on contrastive learning to approximate the classification loss, adding graph contrastive learning approaches as baselines. The authors did a nice job addressing reviewers’ comments and adding suggested experiments. Overall, the reviewers have positive assessment after author responses.

**Note From Pc:**

if the above contains the word "oral" or "spotlight" please see: "oral" presentation means -> notable-top-5% and "spotlight" means -> notable-top-25%. As stated in our emails, we are disassociating presentation type from AC recommendations